# A bicycle can be balanced by stochastic optimal feedback control but only with accurate speed estimates

**Eric Maris** *

Donders Institute for Brain, Cognition and Behaviour, Radboud University, Nijmegen, The Netherlands

* eric.maris@donders.ru.nl

**Data Availability Statement:** All relevant data are within the paper and its Supporting Information files.

**Funding:** The authors received no specific funding for this work.

## Abstract

Balancing a bicycle is typical for the balance control humans perform as a part of a whole range of behaviors (walking, running, skating, skiing, etc.). This paper presents a general model of balance control and applies it to the balancing of a bicycle. Balance control has both a physics (mechanics) and a neurobiological component. The physics component pertains to the laws that govern the movements of the rider and his bicycle, and the neurobiological component pertains to the mechanisms via which the central nervous system (CNS) uses these laws for balance control. This paper presents a computational model of this neurobiological component, based on the theory of stochastic optimal feedback control (OFC). The central concept in this model is a computational system, implemented in the CNS, that controls a mechanical system outside the CNS. This computational system uses an internal model to calculate optimal control actions as specified by the theory of stochastic OFC. For the computational model to be plausible, it must be robust to at least two inevitable inaccuracies: (1) model parameters that the CNS learns slowly from interactions with the CNS-attached body and bicycle (i.e., the internal noise covariance matrices), and (2) model parameters that depend on unreliable sensory input (i.e., movement speed). By means of simulations, I demonstrate that this model can balance a bicycle under realistic conditions and is robust to inaccuracies in the learned sensorimotor noise characteristics. However, the model is not robust to inaccuracies in the movement speed estimates. This has important implications for the plausibility of stochastic OFC as a model for motor control.

## Introduction

Keeping balance is an important function for many organisms. With this function, the organism controls one body axis relative to gravity, and it achieves this by keeping the body's center of mass (CoG) above its area of support (AoS). In this paper, I will focus on balancing a bicycle. However, much of what I will say also holds for other forms of balancing that involve a human body: walking, running, skating, skiing, etc. For example, all forms of balancing a human body involve the same two basic actions for keeping the body's CoG above its AoS: (1) changing the AoS while keeping the CoG fixed (e.g., by stepping out with one leg), and (2) shifting the CoG

**Competing interests:** The authors have declared that no competing interests exist.

while keeping the AoS fixed (e.g., by leaning the upper body). Keeping balance during walking is an active field of research, and the recent review paper by Bruijn and Van Dieën [1] gives a good overview.

Keeping balance is a sensorimotor control problem: the central nervous system (CNS) receives sensory information about the body, the body-attached tools (bicycle, skates, skis, . . .), and their environment (turn radius, speed, . . .), and uses this information for calculating actions with which it controls the position of body and tools relative to gravity. The dominant model for sensorimotor control assumes that the CNS makes use of an internal model to determine these control actions [2, 3]. In some publications [4, 5], a distinction is made between forward and inverse internal models, but here I will only consider forward models. The (forward) internal model simulates the dynamics of the plant (body plus body-attached tools) it attempts to control.

A very influential version of this model claims that this control is optimal in the sense that it minimizes a cost functional that depends on movement precision (here, deviation from the upright position) and energetic costs [2, 3]. This model is called optimal feedback control (OFC), and in this paper I will apply the model's stochastic version to bicycle balance control; the deterministic version would predict that the CoG stays exactly above the AoS once this position is reached, which is unrealistic.

To evaluate the plausibility of stochastic OFC as a model for bicycle balance control, one must address at least the following questions: (1) Is the model good enough to balance a bicycle under realistic conditions (lean and steering angles that are observed with real riders), and (2) Is the model robust to inaccuracies in the model parameters? The relevance of robustness follows from the fact that the model parameters must allow for an accurate simulation of the plant dynamics. However, in some inevitable cases (e.g., in the beginning of a learning process), the parameter values cannot be very close to their optimal values, and therefore the model must have some minimal robustness to inaccurate parameter values. Of course, the stabilization performance (indexed by, e.g., lean angle variability) may decrease with parameter inaccuracy, but for a realistic range of values (see *Results*), the bicycle and rider should not fall over.

It is useful to distinguish different types of inaccuracies with respect to the time it takes to reduce them. On the one extreme, there are inaccuracies that are reduced between (instead of within) cycling trips. These inaccuracies pertain to slowly varying characteristics of the plant (gain factors, moment arms, sensorimotor noise characteristics, . . .) that the CNS must learn from experience.

At the other extreme, there are the inaccuracies in the state variables (i.e., the variables of the equations of motion). In this paper, all state variables are related to limb configurations and gravity (steering angle, upper- and lower body angle) about which the CNS obtains information via the somatosensory (including proprioception) and the vestibular system. Inaccuracies in the CNS-estimated state variables are reduced on a timescale that is set by the delays in these sensory systems, which are around 100 ms [6]. Although I will not investigate this in the present paper, some minimal robustness is required to inaccuracies in these estimates. Fortunately, there is good evidence from psychophysical studies that, in healthy humans, the CNS obtains reliable sensory information about the body's orientation relative to gravity: for body orientations near the vertical axis, the noise standard deviation of the CNS's estimate is approximately 4 degrees [7].

In between these two extreme time scales (slowly varying plant characteristics and state variables), there is an intermediate time scale that is characteristic for parameters such as movement speed. According to the literature, movement speed estimates depend on optical flow [8]. However, these estimates are very unreliable, as is clear from its Weber fraction (the

smallest step increase in forward optical flow velocity necessary for the difference to be perceived): to perceive an increase within 500 ms. the increase had to be at least 50% [9]. Therefore, a plausible model for bicycle balance control must have some minimal robustness to inaccuracies in movement speed estimates.

In the remainder of this introduction, I will first describe the mechanical aspects of bicycle balance control, and next how bicycle balance control can be formulated as a stochastic optimal control problem. In the Results section, I will first introduce a model of sensorimotor control that is based on the idea that a mechanical system (plant) is both controlled and learned by a computational system that uses an internal model to calculate optimal control actions. Next, in a simulation study, I will evaluate (1) whether this model is good enough to balance a bicycle under realistic conditions, and (2) whether it is robust to inaccuracies in the values of two parameters of the computational system, sensorimotor noise characteristics (slowly varying) and movement speed (intermediate time scale).

## Control actions for balancing a bicycle

**Problem definition.** A standing human is balanced when his center of gravity (CoG) is above his area of support (AoS), which is formed by the soles of his two feet plus the area in between. Balance follows from the fact that the gravitational force (a vector quantity in 3D passing through the CoG) intersects this AoS. The situation is similar but not identical for a bicycle. A stationary bicycle (i.e., a bicycle in a track stand) is balanced when the combined CoG of rider and bicycle is above the one-dimensional line of support (LoS), the line that connects the contact points of the two wheels with the road surface. In this position, the direction of the gravitational force intersects the LoS. However, because of disturbances, the CoG cannot be exactly above this one-dimensional LoS for a finite period. Therefore, a bicycle is considered balanced if the CoG fluctuates around the LoS within a limited range, small enough to prevent the bicycle from touching the road surface.

Compared to a stationary bicycle, the balance of a moving bicycle is more complicated because, besides gravity, also the centrifugal force acts on the CoG. Crucially, the centrifugal force is under the rider's control via the turn radius [10]. The balance of a moving bicycle depends on the resultant of all forces that act on the CoG: a bicycle is balanced if the direction of this resultant force fluctuates around the LoS within a fixed range. Besides the forces that act on the CoG, there are also forces that are responsible for the turning of the bicycle's front frame, and some of these do not depend on the rider [11]. These latter forces are responsible for the bicycle's self-stability and will be discussed later (see *Bicycle self-stability*).

**The geometry of the rider-bicycle combination.** The control actions with which a rider can balance his bicycle are constrained by the geometry of the bicycle and the rider's position on it. To describe the possible control actions, I start from the kinematic variables of a model of the rider-bicycle combination, shown in Fig 1. This model consists of three rigid bodies: front frame, rear frame, and the rider's upper body. The positions of these three bodies are specified by three angular variables: (1) the steering angle (the position of the front frame relative to the rear frame), denoted by $\delta$, (2) the rear frame lean angle (the position of the rear frame relative to gravity), denoted by $\phi_1$, and (3) the upper body lean angle (the position of the upper body relative to gravity), denoted by $\phi_2$.

I assume that the rider sits on the saddle and keeps his feet resting on the non-moving pedals. In this position, the rider's lower body (the hips/pelvis and below) is firmly supported and can be considered a part of the rear frame. This simplification implies that leg movements are not used for balance control. However, the stochastic version of this bicycle model (see, *Sensorimotor noise and stochastic OFC*) allows for pedaling-related movements to be included as

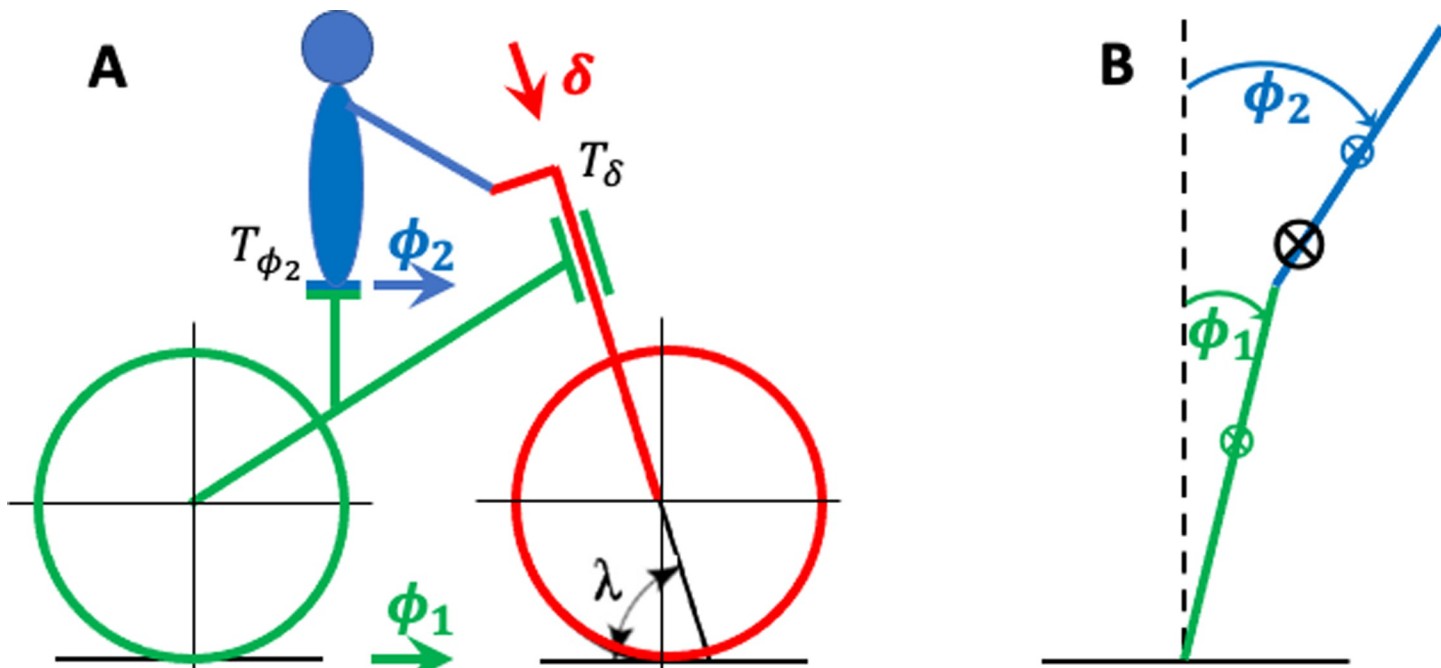

**Fig 1. Kinematic variables of the bicycle model plus the rider-controlled torques.** (A) Side view. In green, the bicycle rear frame, characterized by its lean angle $\phi_1$ over the roll axis (green arrow). In red, the bicycle front frame, characterized by its angle $\delta$ over the steering axis (red arrow). In blue, the rider's upper body, characterized by its lean angle $\phi_2$ over the roll axis (blue arrow). In black, (1) the steering torque $T_\delta$ and the lean torque $T_{\phi_2}$, which are both applied by the rider, and (2) the steering axis angle $\lambda$, which is set equal to 90 degrees for the purposes of the present paper (see text). (B) Rear view. In green, the bicycle rear frame (plus lower body) lean angle $\phi_1$. In blue, the rider's upper body lean angle $\phi_2$. The symbol $\otimes$ denotes the CoG of the upper body (in blue), the lower body (in green), and the combined CoG (in black).

motor noise and, in this way, to produce associations between lean- and steering angles. In sum, the only forces that can be used for balance control are (1) a steering torque $T_\delta$ on the handlebars, and (2) a lean torque $T_{\phi_2}$ at the hinge between the rider's upper body and the rear frame, corresponding to the hips.

**Cycling involves a double balance problem.** Cycling involves a double balance problem, of which I have described the first part: keeping the combined CoG of the rider and the bicycle above the LoS. The second balance problem pertains to the rider only: keeping his CoG above his AoS. In the bicycle model in Fig 1, the rider is represented by his upper body, of which the CoG must be kept above the saddle. The balance problem with respect to the rider is further simplified by only considering the balance over the roll axis (parallel to the LoS), which corresponds to upper body movements to the left and the right. I thus ignore the balance over the pitch axis (perpendicular to the LoS and gravity), which corresponds to upper body movements to the front and the back, typically caused by accelerations and braking. With this simplification, the joint between the rider's upper body and the rear frame is a hinge with a single degree of freedom.

**Balance control strategies from a mechanical point of view.** For keeping the combined CoG over the LoS (the first balance problem), the relevant control actions must result in a torque over the LoS (roll axis). Within the constraints of our kinematic model, there are two ways for a rider to perform a control action: (1) by turning the handlebars, and (2) by leaning the upper body. To explain these control actions, it is convenient to make use of Fig 1B. This is the schematic of a double compound pendulum, of which the dynamics depend on how it is actuated: (1) if the contact between the green rod and the road surface is controlled by a linear

force, the dynamics is known as the "double compound pendulum on a cart" [12], and (2) if the angle between the green and the blue rod (the upper body lean angle) is controlled by a torque at this joint, the dynamics is known as the Acrobot [13].

We first take the perspective of a double compound pendulum on a cart. This involves that, by turning the handlebars, the contact point of the green rod (representing the combined front and rear frame) with the road surface moves to the right under the combined CoG. In fact, turning the handlebars changes the trajectory of the tire-road contact points and, because the CoG wants to continue in its pre-turn direction (by Newton's first law), this results in a centrifugal force in the bicycle reference frame (of which the LoS is one of the defining axes). This centrifugal force is perpendicular to the LoS and results in a torque over the roll axis in the direction opposite to the turn (a tipping out torque). This steering-induced tipping out torque can be used to move the combined CoG to the opposite side of the turn. Thus, steering in the direction of the lean produces a tipping out torque that brings the combined CoG over the LoS. This explains the name of this control mechanism: "steering to the lean".

We now take the perspective of the Acrobot, which involves that, by applying a lean torque at the hips, the lean angles of both body parts change. Consequently, the separate CoGs of both body parts are shifted, and this in turn affects the gravity-dependent torques on these body parts. Crucially, a lean torque at the hips does not shift the combined CoG, and therefore cannot bring this combined CoG above the LoS in a direct way. However, it can do so in an indirect way, namely by turning the front frame. This is essential for the mechanism via which a bicycle can be balanced when riding no handed. First, when leaning the upper body sufficiently to one side, the bicycle and the lower body lean to the other side. Next, depending on properties of the bicycle (wheel flop, the wheels' gyroscopic forces, the combined CoG [11, 14]), leaning the bicycle to one side turns the front frame to the same side. This lean-induced turn of the front frame then initiates the same mechanism as when turning the front frame by means of the handlebars: a change in the trajectory of the tire-road contact points results in a centrifugal force perpendicular to the LoS, producing a torque over the roll axis in the direction opposite to the turn. This lean torque brings the LoS under the CoG.

For the second balance problem (keeping the upper body's CoG over its AoS), the same two control actions can be used: (1) turning the handlebars, and (2) applying a lean torque at the hips. Turning the handlebars in the direction of the upper body lean produces a lean torque in the other direction (i.e., away from the lean), and this allows to control this upper body lean angle. By applying a lean torque at the hips, this upper body lean angle can be controlled in a more direct way, but at the expense of leaning the bicycle (and the lower body) in the opposite direction.

Because the two balance problems use the same control actions, coordination is required. For example, a torque at the hips can be used to counter the turning-induced centrifugal torque on the upper body: by applying a hip torque of equal magnitude as this centrifugal torque (but opposite direction), the position of the upper body can be controlled. There exists an energy-efficient alternative for this upper body control strategy, well-known in motorcycle racing: leaning the upper body to the inside of the turn. When the upper body is sufficiently leaned to the inside of the turn, the resulting gravity-induced torque will counter the centrifugal torque on the upper body CoG.

**Balancing and steering.**   When riding a bicycle, the rider typically does not only want to balance his bicycle, but also wants to steer it over a chosen/indicated trajectory. This paper only considers control actions for balancing the bicycle, and therefore will not consider constraints on the trajectory, such as obstacles and bicycle path edges. This pure balance task corresponds to cycling blindfolded on an empty parking lot. After a brief familiarization, most humans can cycle blindfolded on an empty parking lot; a search on social media will show several demonstrations of this.

**Bicycle self-stability.** At this point, it is necessary to mention the self-stability of the bicycle, the fact that, within some range of speeds, the bicycle is balanced without control actions by the rider [11]. Self-stability is investigated by modelling the rider as a mass that is rigidly attached to the rear frame and does not touch the front frame, allowing the handlebars to move freely. Self-stability depends on multiple factors, such as geometric trail, pneumatic trail, wheel flop, the wheels' gyroscopic forces, and the combined CoG [11, 14]. These factors all contribute to the bicycle's tendency to steer in the direction of the lean.

The focus of the present paper is on the rider's contribution to bicycle stability, and therefore I used a model bicycle from which I removed all known factors that contribute to the bicycle's self-stability (see Fig 2). Specifically, I removed the effects of pneumatic trail and the wheel's gyroscopic forces by replacing the wheels by ice skates (or, equivalently, tiny roller skate wheels). And I removed the effects of geometric trail and wheel flop by choosing a vertical steering axis (i.e., by setting λ in Fig 1 to 90 degrees), as in most bicycles for bicycle moto-cross (BMX) and artistic cycling. I also keep the CoG at approximately the same position as on a regular bicycle (i.e., 30 cm before the rear wheel contact point), because a CoG with a more anterior position may result in bicycle self-stability [14]. Without all these effects, the bicycle's front frame does not steer in the direction of the lean unless the rider turns the handlebars. Therefore, the balance control strategy for riding no handed that I described before, cannot be used on this model bicycle. In the Methods and Models section, I will describe how this simplified bicycle-rider combination can be modeled as a double pendulum of which the base can be moved by turning the front wheel/skate, and the joint at the hips can be actuated. This model will be called the "steered double pendulum" (SDP).

**Linear and nonlinear bicycle models.** The SDP is a nonlinear model. This nonlinearity a desirable property because the objective of the present paper is to demonstrate that a linear control mechanism can balance a nonlinear mechanical system. The most popular bicycle model is linear, and it was proposed by Meijaard, Papadopoulos et al. [11] as a benchmark for studying the passive dynamics of a bicycle. Depending on the model parameters, this linear model is self-stable in some range of speeds.

For comparison with the nonlinear SDP without self-stabilizing forces, I also used a linear bicycle model *with* self-stabilizing forces (gyroscopic forces plus the forces that depend on

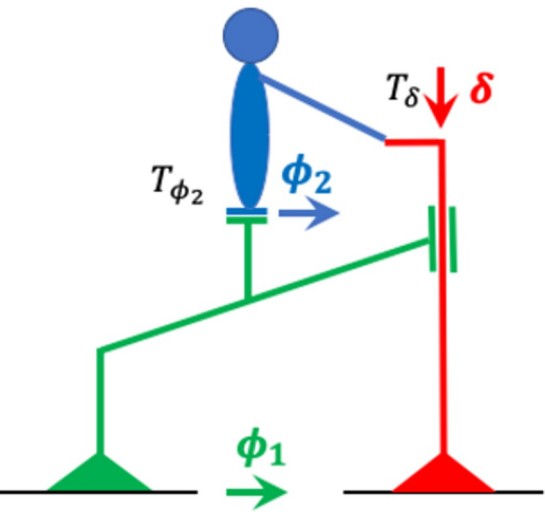

**Fig 2. Bicycle model without the known factors that affect bicycle self-stability.** Compared to Fig 1, this model has ice skates instead of wheels and a vertical steering axis.

geometric trail and wheel flop) to check the generality of some results obtained with the nonlinear SDP without self-stabilizing forces. This linear model will be called the "benchmark double pendulum" (BDP), and it is a combination of an existing benchmark model [11] and the double pendulum.

Another nonlinear bicycle model has been proposed by Basu-Mandal, Chatterjee et al. [15]. Just like the linear benchmark model [11], this nonlinear model does not include an upper body. However, it turns out to be straightforward to produce the BDP by linearizing the double pendulum and combining these with the linear benchmark model. Such an extension was much less straightforward starting from the nonlinear model by Basu-Mandal, Chatterjee et al. [15]. This motivates my choice for the SDP and the BDP. For this paper, the mechanical details of the bicycle model are not crucial. Instead, it is my objective to demonstrate that a linear control mechanism can balance a whole range of bicycle models and does so under realistic conditions. Because almost all mechanical systems are nonlinear, I will focus on the SDP, and use the BDP to investigate the generality of the SDP results.

**Control and noise forces.** For investigating balance control, one must distinguish between control and noise forces. Loosely formulated, control forces are the forces that the rider uses to balance the bicycle. For a more precise formulation, I use the optimal control framework, which defines control actions as the actions that optimize a quantitative performance index. Thus, control forces are the optimal forces for a given performance index.

Noise forces are the difference between the forces that are applied and the optimal control forces. It is useful to distinguish between (1) noise forces that originate from the rider, and (2) noise forces that originate from interactions of the bicycle with the environment (e.g., collisions, gusts of wind). In this paper, I only consider noise forces that originate from the rider. These noise forces affect the balance via the same contact points as the two control forces (the handlebars and the saddle). These noise forces are an important instrument in the simulations that I have run to investigate bicycle balance control: they distort the balance, and this allows to investigate different stabilizing (balance-restoring) mechanisms.

## Balancing a bicycle as a stochastic optimal control problem

**Optimal feedback control.** Every motor task can be performed in an infinite number of ways, and this is for two reasons: (1) the human body has a very large number of joints that can be used in various combinations to produce the same trajectory of the relevant body part (the combined CoG in a balance task, an effector endpoint in reaching task, . . .), and (2) a motor task unfolds over time and can be performed with different speed profiles. Nevertheless, most motor tasks are performed in a highly stereotyped manner. For instance, reaching tasks consistently show roughly straight-line paths with bell-shaped speed profiles [3].

To explain these highly stereotyped actions among skilled performers, Todorov and colleagues [3, 16] proposed optimal feedback control (OFC). This theory uses a scalar cost functional that increases with time-integrated imprecisions and energetic costs. OFC involves that the control actions are chosen such that this cost functional is minimized. OFC is sometimes proposed as an alternative for feedforward trajectory planning [2], but this incorrectly suggests that OFC and trajectory planning cannot be combined. In this paper, I will present a model for the specific task of balance control, as when cycling blindfolded on an empty parking lot. For this specific task, one can ignore all aspects of cycling that involve trajectory planning, such as steering a bicycle over an indicated path or an obstacle course.

In previous work, OFC has been mainly applied to reaching tasks [2, 16–19]. For such tasks, the overall precision predominantly depends on the precision at the endpoint of the reaching movement. In line with this fact, the cost functional is dominated by imprecisions

(distances between the end effector and the reach target) near the endpoint [16]. In contrast, for tasks in which a state must be maintained over time, such as balancing a bicycle, the cost functional must depend on the imprecisions uniformly across the theoretically infinite lean angle trajectory.

For applying OFC, one needs the equations of motion (EoM) that describe the dynamics of the system (here, the rider-bicycle combination) as a set of differential equations. The variables of these differential equations are called state variables, and for my two bicycle models (SDP and BDP) they are the following: the steering angle $\delta$, the rear frame lean angle $\theta_1$, the upper body lean angle $\theta_2$ (see Fig 2), plus their corresponding angular rates. In the *Materials and Methods*, I will derive the SDP EoM from Lagrangian mechanics, and the BDP EoM by linearizing the double pendulum EoM and combining these with the linear benchmark model.

The fact that the SDP EoM are nonlinear has important implications for the use of OFC for stabilization. Specifically, OFC does not provide general results for stabilizing a nonlinear system. However, it provides very useful results for stabilizing a linear system, and this has led to the common practice in robotics to linearize the nonlinear system, apply OFC for linear systems, and use the resulting optimal control signals to stabilize the nonlinear system [13]. I hypothesize that the CNS implements a similar solution for stabilizing a bicycle and the rider's upper body: build an internal linear approximation of the external nonlinear system that the CNS wants to stabilize and use calculations like those from OFC to achieve this. In a later section, *Stabilizing a nonlinear mechanical system by linear stochastic OFC*, I will describe this model in more detail.

OFC uses a scalar cost functional to define the optimal control actions. This is in line with the fact that the CNS implements functions for setting goals and evaluating actions. For our application to bicycle balance control, it is natural to define this cost functional as one that increases with (1) deviations between the CoGs (combined and upper body) and their respective support, and (2) the energetic costs of the control actions. The control actions that result from the minimization of this cost functional are the steering and the lean torque.

**Sensorimotor noise and stochastic OFC.**   Because riders and other biological systems suffer from sensor and motor noise [20], deterministic OFC is an unrealistic model for bicycle balance control. As a result of this noise, the CNS cannot perfectly know nor control the outside world, which includes the body that is attached to the CNS. Specifically, if the sensory feedback is noisy, the CNS cannot infer the state variables perfectly from this feedback. Also, the CNS is unaware of the motor noise that is generated at the muscular level, which is added after the CNS has produced the motor command. Therefore, even if the CNS were able to calculate an optimal motor command based on perfectly accurate state information, that command would not fully control the muscles.

Fortunately, for a system whose behavior depends on noise, optimal control is still defined, namely if the system is governed by linear stochastic differential equations (SDEs) with additive Gaussian noise and a quadratic cost functional. Under these conditions, control is optimal if it is based on an optimal state estimate [21]. The optimality of this state estimate is relative to the conditional probability distribution of the state estimate at time $t$ given the values of all variables on previous times. Therefore, this optimal estimate not only depends on the sensory feedback at time $t$, but also on the optimal state estimate and the control action (actually, its efference copy) just before this time. This optimal estimate involves the familiar Kalman filter, which weights the sensory feedback in proportion to its reliability. Several empirical studies have suggested that state estimation in the CNS involves this type of weighting in proportion to the reliability of the available information [22–24].

The ability to correct for motor and sensor noise depends on the CNS's internal model of the dynamics of the plant and the sensory feedback. The CNS uses this internal model to

estimate the current state from (1) the previous state, (2) the most recent control action, and (3) the sensory feedback. Several psychophysical [25–27] and neurophysiological [28, 29] studies have provided evidence for such internal models. An internal model can be conceived as a set of differential equations that allows the CNS to simulate state variables and to combine this information with the sensory feedback to obtain an optimal state estimate.

**The robustness of control based on an internal model.** Because an internal model cannot be directly observed, its hypothesized role in sensorimotor control must be evaluated based on its performance. This performance pertains to how well the optimal controls under a linear approximation can stabilize a nonlinear system. This linear approximation involves several parameters, such as the matrices that define the linear approximation to the nonlinear EoM and the noise covariance matrices (see *Stabilizing a nonlinear mechanical system by linear stochastic OFC*). The larger the range of parameter and state values for which the internal model can stabilize the nonlinear system, the more robust the control. Because the CNS must learn the internal model from experience with the mechanical system, which may be a slow process, the imperfect internal must be robust to some inaccuracies in the internal model.

In this paper, for two types of parameters, the learned sensorimotor noise characteristics, and movement speed, I will determine the range of values for which the internal model can stabilize the nonlinear bicycle model while producing realistic state values. From these values, it can be concluded that the model is robust to inaccuracies in the learned sensorimotor noise characteristics, but not to inaccuracies in the movement speed estimates.

## Methods and models

### Equations of motion (EoM) for the steered double pendulum (SDP)

The SDP is depicted schematically in Fig 1. The SDP contains ingredients of three familiar models: the double compound pendulum on a cart [DCPC, 12], the Acrobot [13], and the torsional spring-mass-damper system. Roughly speaking, the SDP is a double compound pendulum of which the base can be steered by a wheel (instead of a cart) and the joint between the two rods (at the hips) can be actuated, as in the Acrobot. Both actuated joints, one at the handlebars and one at the hips, are modeled as a torsional spring-mass-damper system. I will denote the lower and the upper rod as, respectively, the lower and the upper body. The lower body represents the rear frame plus the rider's lower body; the upper body only represents the rider's upper body.

**The kinematic model.** Fig 3 depicts the relevant kinematic variables in both an inertial (yellow origin) and a rider/bicycle-centered (purple origin) reference frame. The inertial reference frame has an arbitrary origin, a vertical coordinate axis V perpendicular to gravity, and an arbitrary horizontal coordinate axis H perpendicular to V. The rider/bicycle-centered reference frame has its origin at the orthogonal projection of the combined CoG on the LoS, and a vertical and horizontal coordinate axis V' and H' that are parallel to those of the inertial reference frame. The rider/bicycle-centered reference frame is non-inertial because, when the bicycle turns, the origin no longer moves in a straight line, and therefore accelerates in the inertial reference frame.

I will use the rider/bicycle-centered reference frame to define three kinematic variables. The first two kinematic variables are the lower and the upper body lean angles ($\phi_1$ and $\phi_2$), which are defined relative to the vertical axis V'. The third kinematic variable is the yaw angle $\psi$ of the LoS, which is defined relative to the horizontal axis V'. When describing the dynamics of the SDP, we need an expression for the centrifugal acceleration $\alpha$ at the combined CoG. I assume identical speeds at the separate CoGs of the lower and the upper body as well as identical angular rates of the projections on the horizontal plane. Then, the centrifugal acceleration

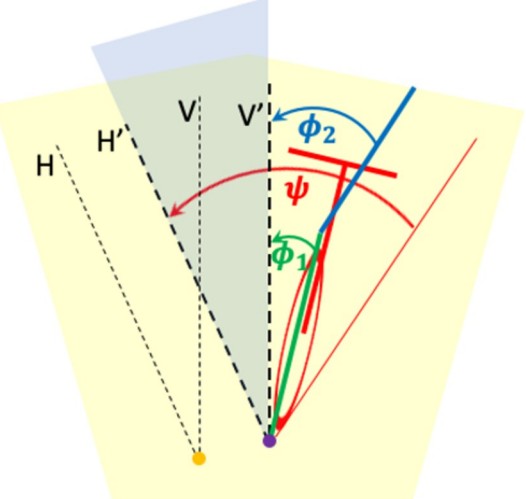

**Fig 3. The relevant kinematic variables of the SPD in both an inertial (yellow origin) and a rider/bicycle-centered (purple origin) reference frame.** The inertial reference frame has an arbitrary origin, and the rider/bicycle-centered reference frame has its origin at the orthogonal projection of the combined CoG on the LoS. These reference frames have parallel coordinate axes. In green and blue, I depict the lean angles of the lower and the upper body ($\phi_1$ and $\phi_2$), and in red, I depict the yaw angle $\psi$ of the LoS. The horizontal plane (road surface) is colored light yellow.

only depends on the yaw angular rate $\dot{\psi} = \partial\psi/\partial t$ and the speed $v$:

$$\alpha = v\dot{\psi}$$

Crucially, $\dot{\psi}$ depends on the steering angle $\delta$, and this allows the rider to control the LoS.

For the SDP EoM, one must know the precise dependence of $\dot{\psi}$ on $\delta$. Deriving this dependence is a well-known problem in vehicle dynamics [30], and here I use the known result. This result involves the so-called slip angle $\beta(\delta)$, which is the angle between the velocity vector of the combined CoG and the LoS. This slip angle can be obtained as follows:

$$\beta(\delta) = \tan^{-1}\left(\frac{w_r \tan(\delta)}{W}\right)$$

In this equation, $W$ is the wheelbase and $w_r$ is the position of the combined CoG on the LoS. More precisely, $w_r$ is the distance between the road contact point of the rear wheel and the orthogonal projection of the combined CoG on the LoS. For realistic values ($W = 1.02$, $w_r = 0.3$, -20˚ $< \delta <$ 20˚), the slip angle $\beta(\delta)$ is almost a linear function of $\delta$:

$$\beta(\delta) \approx \frac{w_r\delta}{W}$$

For steering angles -20˚ $< \delta <$ 20˚, all deviations from linearity are less than 0.36%. I will continue to use this approximation. Following [30], one can obtain the centrifugal acceleration $\alpha(\delta)$ as follows:

$$\alpha(\delta) = v^2 \frac{\cos(\beta(\delta))}{W}\tan(\delta) \qquad\qquad \text{Eq 1}$$

For a constant speed $v$, the centrifugal acceleration is only a function of the steering angle $\delta$.

**The steering model.**   The steering model assumes that the steering angle $\delta$ is fully controlled by rider-applied forces on the handlebars. Thus, I ignore all forces that may contribute to the bicycle's self-stability.

The steering assembly consists of the front wheel, the fork, the handlebars, and the rider's arms. I model this assembly as a torsional spring-mass-damper system:

$$I_{steer}\,\ddot{\delta} + C_{steer}\,\dot{\delta} + K_{steer}\delta = T_\delta \qquad\qquad \text{Eq 2}$$

In Eq 2, $I_{steer}$ is the assembly's rotational inertia, $C_{steer}$ its damping, and $K_{steer}$ its stiffness. The input to the steering assembly is the net torque produced by the rider's muscles and denoted by $T_\delta$ on the right-hand side of Eq 2.

**The double compound pendulum with a steer-actuated base.**   I model the lean angles $\phi_1$ and $\phi_2$ as the result of a double compound pendulum on a virtual (zero mass) cart with acceleration equal to $\alpha(\delta)$, the centrifugal acceleration derived under our kinematic model (see Eq 1). Like the Acrobot, this double compound pendulum has an actuated joint between the upper and the lower body (the hips). To make the model more biologically realistic, I add stiffness and damping to the hips.

The EoM for $\phi_1$ and $\phi_2$ are obtained by first applying the Euler-Lagrange method to the DCPC with a zero-mass cart, and then adding the constraint that the cart is controlled by the steering-induced centrifugal acceleration $\alpha(\delta)$. The derivation of the DCPC EoM using the Euler-Lagrange method can be found in the literature. Here, I started from Bogdanov (12) and added stiffness, damping and torque input at the joint between the two rods (the hips). Next, I added the constraint that the angles $\phi_1$ and $\phi_2$ have no direct effect on the position of the base of the first rod (in the DCPC, the point where the cart is attached). This constraint follows from the fact that the bicycle's wheels are oriented perpendicular to the cart's wheels. Under this constraint, the position of the base of the first rod is fully controlled by the steering-induced centrifugal acceleration $\alpha(\delta)$. The result is the following:

$$
\begin{bmatrix} d_1\cos(\phi_1) \\ d_2\cos(\phi_2) \end{bmatrix}\alpha(\delta) +
\begin{bmatrix} d_3 & d_4\cos(\phi_1-\phi_2) \\ d_4\cos(\phi_1-\phi_2) & d_5 \end{bmatrix}
\begin{bmatrix} \ddot{\phi}_1 \\ \ddot{\phi}_2 \end{bmatrix}
$$
$$
+\begin{bmatrix} 0 & d_4\sin(\phi_1-\phi_2)\phi_2 \\ d_4\sin(\phi_1-\phi_2)\phi_1 & 0 \end{bmatrix}
\begin{bmatrix} \dot{\phi}_1 \\ \dot{\phi}_2 \end{bmatrix}
$$
$$
+\begin{bmatrix} -f_1\sin(\phi_1) \\ -f_2\sin(\phi_2) \end{bmatrix}
$$
$$
+\begin{bmatrix} K_{pelvis}(\phi_1-\phi_2) + C_{pelvis}(\dot{\phi}_1-\dot{\phi}_2) \\ -K_{pelvis}(\phi_1-\phi_2) - C_{pelvis}(\dot{\phi}_1-\dot{\phi}_2) \end{bmatrix} =
\begin{bmatrix} 0 \\ T_\phi \end{bmatrix}
\qquad \text{Eq 3}
$$

The crucial difference between Eq 3 and the corresponding equation for the DCPC is that $\alpha$ ($\delta$) replaces the acceleration of the cart. The constants in Eq 3 are defined as follows:

$$
\begin{aligned}
d_1 &= m_1 l_1 + m_2 L_1 \\
d_2 &= m_2 l_2 \\
d_3 &= m_1 l_1{}^2 + m_2 L_1{}^2 + I_1 \\
d_4 &= m_2 L_1 l_2 \\
d_5 &= m_2 l_2{}^2 + I_2 \\
f_1 &= (m_1 l_1 + m_2 L_1) g \\
f_2 &= m_2 l_2 g
\end{aligned}
\qquad \text{Eq 4}
$$

The constants $m_1$, $L_1$, $l_1$ and $I_1$ are, respectively, the mass, the length, the CoG ($L_1/2$) and the mass moment of inertia of the double pendulum's first rod, which represents the bicycle and the rider's lower body. The constants $m_2$, $L_2$, $l_2$ and $I_2$ are defined in the same way, but now for the second rod, which represents the rider's upper body. Further, $g$ is the gravitational constant, and $K_{pelvis}$, $C_{pelvis}$ and $T_\phi$ are the stiffness, the damping, and the torque at the hips.

The SDP EoM are obtained from Eqs 2 and 3 by deriving expressions for the second derivatives $\ddot\delta$ and $\left[\ddot\phi_1, \ddot\phi_2\right]^T$. These expressions are complicated and not insightful. I use these EoM to define the state-space equations $\dot{x} = \Omega(x, u + m)$ for the state variables $x = \left[\delta, \phi_1, \phi_2, \dot\delta, \dot\phi_1, \dot\phi_2\right]^T$, external forces $u = \left[T_\delta, T_{\phi_2}\right]^T$, and motor noise $m$.

**An optimal linear approximation of the SDP EoM.** In our model for sensorimotor control, the computational system is a linear approximation of $\Omega(x, u)$. I find an optimal linear approximation by calculating the Jacobian of $\Omega(x, u)$ at the unstable fixed point $x = 0$ and without external input (i.e., $u = 0$). I obtained this Jacobian using the Matlab function jacobian.m. By taking the Jacobian of $\Omega(x, u)$ with respect to $x$ and $u$, I obtain, respectively, the matrices $A$ and $B$. This allows for the following approximation near the unstable fixed point:

$$
\dot{x} \approx Ax + Bu
$$

I numerically evaluated the accuracy of this approximation by calculating finite differences $[\Phi(\varepsilon, 0) - \Phi(0, 0)]/\varepsilon$ (for $A$) and $[\Phi(0, \varepsilon) - \Phi(0, 0)]/\varepsilon$ (for $B$) for decreasing values of $\varepsilon$. I found that for $\varepsilon \to 0$ the finite difference approximations converged to $A$ and $B$.

## Equations of motion (EoM) for the benchmark double pendulum (BDP)

The BDP is based on three ideas. The first idea is to follow the approach of Meijaard, Papadopoulos et al. [11] and derive linearized EoM for a bicycle with the rider's lower body rigidly attached to the rear frame and no upper body. These linearized EoM depend on a number of constants, and I chose these constants such that (1) the front frame is as similar as possible to the self-stable benchmark bicycle model described by Meijaard, Papadopoulos et al. [11], and (2) the lengths and masses are as similar as possible to the SDP. The second idea is to model the interactions between the upper body and the rear frame (which includes the lower body) by the linearized EoM of the double compound pendulum, similar to Dialynas, Christoforidis et al. [31]. The nonlinear EoM of the double compound pendulum are obtained from Eq 3 by removing the terms that correspond to the centrifugal acceleration $\alpha(\delta)$, the stiffness and the damping. Finally, the third idea is to first derive the BDP EoM without stiffness and damping terms, and to add these terms only in the last step.

The approach of Meijaard, Papadopoulos et al. [11] involves a method to calculate the defining matrices of linearized EoM of the following type:

$$
\boldsymbol{M}\begin{bmatrix}\ddot{\delta}\\\ddot{\phi}_1\\\ddot{\phi}_2\end{bmatrix} + \boldsymbol{C}\begin{bmatrix}\dot{\delta}\\\dot{\phi}_1\\\dot{\phi}_2\end{bmatrix} + \boldsymbol{K}\begin{bmatrix}\delta\\\phi_1\\\phi_2\end{bmatrix} = 0
$$

The matrices $\boldsymbol{M}$, $\boldsymbol{C}$ and $\boldsymbol{K}$ are functions of several constants (angles, lengths, masses, mass moments of inertia, gravitational acceleration, speed) that characterize the bicycle components and the internal forces that act on them. However, Meijaard, Papadopoulos et al. [11] only derived linearized EoM for bicycles with a rider that was rigidly attached to the rear frame. Thus, the upper body lean angle $\phi_2$ is absent from their EoM. This missing component can be obtained by linearizing the double pendulum EoM which models the interactions between $\phi_1$ and $\phi_2$. Schematically, each of the matrices $\boldsymbol{M}$, $\boldsymbol{C}$ and $\boldsymbol{K}$ is composed as follows:

$$
\begin{bmatrix}MP(1,1) & MP(1,2) & 0\\MP(2,1) & MP(2,2) & 0\\0 & 0 & 0\end{bmatrix} + \begin{bmatrix}0 & 0 & 0\\0 & DP(1,1) & DP(1,2)\\0 & DP(2,1) & DP(2,2)\end{bmatrix}
$$

in which "MP" denotes "Meijaard, Papadopoulos et al" [11], and "DP" denotes "Double Pendulum". The MP calculations were performed by means of the Matlab toolbox Jbike6 [32], in which I entered the constants for a bicycle with the rider's lower body rigidly attached to the rear frame and no upper body. This produced the constants $MP(i,j)$ ($i, j$ = 1,2) for $\boldsymbol{M}$, $\boldsymbol{C}$ and $\boldsymbol{K}$.

I now model the interactions between the upper body and the rear frame by the linearized EoM of the double compound pendulum. The nonlinear EoM of the double compound pendulum are obtained from Eq 3 by removing the terms that correspond to the centrifugal acceleration $\alpha(\delta)$, the stiffness and the damping:

$$
\begin{bmatrix}d_3 & d_4\cos(\phi_1-\phi_2)\\d_4\cos(\phi_1-\phi_2) & d_5\end{bmatrix}\begin{bmatrix}\ddot{\phi}_1\\\ddot{\phi}_2\end{bmatrix}
$$
$$
+ \begin{bmatrix}0 & d_4\sin(\phi_1-\phi_2)\dot{\phi}_2\\d_4\sin(\phi_1-\phi_2)\dot{\phi}_1 & 0\end{bmatrix}\begin{bmatrix}\dot{\phi}_1\\\dot{\phi}_2\end{bmatrix}
$$
$$
+ \begin{bmatrix}-f_1\sin(\phi_1)\\-f_2\sin(\phi_2)\end{bmatrix} = \begin{bmatrix}0\\T_\phi\end{bmatrix}
$$

I evaluate these EoM at $\phi_1 = \phi_2$ and replace $\sin(x)$ by its linear approximation near 0: $\sin(x) \approx x$. This results in

$$
\begin{bmatrix}d_3 & d_4\\d_4 & d_5\end{bmatrix}\begin{bmatrix}\ddot{\phi}_1\\\ddot{\phi}_2\end{bmatrix} + \begin{bmatrix}-f_1 & 0\\0 & -f_2\end{bmatrix}\begin{bmatrix}\phi_1\\\phi_2\end{bmatrix} = \begin{bmatrix}0\\T_\phi\end{bmatrix}
$$

The constants $d_3$, $d_4$ and $d_5$ contain elements that must be added to the matrix $\boldsymbol{M}$, and the constants $f_1$ and $f_2$ contain elements that must be added to the matrix $\boldsymbol{K}$ (for the definitions, see Eq 4). I will use the notation $DP(i, j)$ ($i, j$ = 1,2) to denote these elements. For $\boldsymbol{M}$, the following elements are added:

- $DP(1, 1) = m_2 L_1{}^2$

- DP(1, 2) = DP(2, 1) = $d_4 = m_2 L_1 l_2$

- DP(2, 2) = $d_5 = m_2 l_2^2 + I_2$

And for **K**, the following elements are added:

- DP(1, 1) = $m_2 L_1 g$

- DP(2, 2) = $-f_2 = -m_2 l_2 g$

Finally, I added stiffness and damping terms that were also added to the SDP. The stiffness and damping terms were added to, respectively, **K** and **C**.

## Realistic constants for the SDP and the BDP

I grouped the constants of the two bicycle models in several sets. Within every set, the constants for the SDP are described first, followed by those for the BDP.

**Stiffness, damping and mass moment of inertia for the steering model.**    To assign realistic values to the stiffness and damping parameters of the steering model, it is useful to divide both sides of Eq 2 by $K_{steer}$ and to reparametrize the model as follows:

$$\tau^2 \ddot{\delta} + 2\zeta\tau\dot{\delta} + \delta = \frac{T_\delta}{K_{steer}},$$ 

Eq 5

in which $\zeta$ is the damping ratio and $\tau$ is the time constant. Equating corresponding parts in Eqs 2 and 5, one obtains

$$K_{steer} = \frac{I_{steer}}{\tau^2}$$ 

Eq 6

$$C_{steer} = 2\zeta\tau$$ 

Eq 7

For a damping ratio $\zeta < 1$ the steering assembly oscillates in response to torque input. Because this does not happen in reality, $\zeta$ must be at least 1. The smaller the damping ratio $\zeta$, the faster the response of the steering assembly, which is advantageous for stabilization. I will consider the most responsive steering assembly, and therefore set $\zeta = 1$.

I now set the time constant $\tau$ to an empirically determined value. For that, I make use of the fact that a speeded single joint movement governed by a second order system reaches its maximum speed $\tau$ seconds after the beginning of the movement (see *Empirical determination of the time constant of a critically damped second order system*). From visual inspection of Fig 3B in Lewis & Perreault (2009) [33], I estimate $\tau = 0.33$ seconds. From Eq 7, I find that, in the critically damped case, $C_{steer}$ equals $2\tau$.

The mass moment of inertia $I_{steer}$ has two components, one determined by the bicycle's front assembly ($I_{steer\_bic}$), and one by the rider's arms ($I_{steer\_arms}$). $I_{steer\_bic}$ was calculated as the sum of two component mass moments of inertia: (1) the fork about its main axis, and (2) the wheel about an axis through the rim. These two values were obtained from the MP benchmark model [11]: $I_{steer\_bic} = 0.006 + 0.1405 = 0.1465$.

The mass moment of inertia $I_{steer\_arms}$ results from the fact that the arm muscles must also move themselves plus the bones to turn the front assembly. I treat the arms as 4 kg point masses at the end of the handlebars (turn radius 0.3 m.). It follows that $I_{steer\_arms} = 2 \times 4 \times 0.3^2 = 0.72$ kg m$^2$. Thus,

$$I_{steer} = I_{steer\_bic} + I_{steer\_arms} = 0.1465 + 0.72 = 0.8665 \text{ kg m}^2$$

For the BDP, the complete $3 \times 3$ mass moment of inertia matrix of the front frame must be specified. I specified this matrix using JBike6, in which I adjusted the values of the MP benchmark model. For two of three axes, these values had to be increased by a factor of approximately 12 because the arms were not a part of the MP benchmark model. I specified the stiffness and damping of the front frame in the same way as for the SDP, namely by setting the damping ratio ($\zeta = 1$) and the time constant ($\tau = 0.33$).

**Stiffness, damping and mass moment of inertia for the hips.** I follow the same reasoning as for the steering model, and I also set the damping ratio $\zeta = 1$ and the time constant $\tau = 0.33$. The mass moment of inertia for the hip joint depends on the geometry and the mass of the model for the upper body, which I describe in the next paragraph.

**Lengths and masses of the bicycle and upper body models.** The SDP models the bicycle (plus lower body) and the upper body as rods. I consider a 15 kg. bicycle and a 85 kg. rider with a 45%-55% mass distribution between the lower and the upper body. The bicycle (lower body) height is 1.1 m., and the upper body height is 0.75 m. In terms of the constants in Eq 4:

$$m_1 = (0.45 \times 85) + 15 = 53 \text{ kg}$$

$$m_2 = 0.55 \times 85 = 47 \text{ kg}$$

$$L_1 = 1.1 \text{ m}$$

$$L_2 = 0.75 \text{ m}$$

Using the formula for the mass moment of inertia of a homogeneous rod, I obtain

$$I_1 = \frac{m_1 L_1^{\,2}}{12} = 5.34 \text{ kg m}^2$$

$$I_2 = \frac{m_2 L_2^{\,2}}{12} = 2.2031 \text{ kg m}^2$$

For the BDP rear frame, I adjusted the values of the MP benchmark model to consider the lower mass and CoG. The new values were approximately 75 percent lower than the MP benchmark model.

**Bicycle geometry.** The bicycle geometry parameters were identical to those of the MP benchmark model. Specifically, the wheelbase and the CoG on the LoS were, respectively, $W = 1.02$ and $w_r = 0.3$ m. The angle of steering axis (only relevant for the BDP) was $\lambda = 72$ degrees.

**Gravity and speed.** I set the gravitational constant $g = 9.81$ m/sec$^2$, and the bicycle speed $v = 4.3$ m/sec, the average bicycle speed in Kopenhagen [34].

**Self-stability of the BDP.** Without the upper body, the BDP EoM are for a bicycle with the rider's lower body rigidly attached to the rear frame, and no components taken from the double pendulum EoM. The self-stability of this simplified bicycle can be investigated using the established criterium that the eigenvalues' real parts must be negative. With the constants used in this paper, this simplified bicycle is not self-stable. (This holds with or without the stiffness and damping terms for the arms and the hips.) However, by changing some constants (e.g., the front frame's mass moment of inertia), it is easy to make this simplified bicycle self-stable. I did not do this because I wanted to stay as close as possible to both the MP benchmark model and the SDP.

### Empirical determination of the time constant of a critically damped second order system

I will now show that the time constant $\tau$ of a critically damped second order system can be determined empirically from an experiment in which participants make speeded movements of the joint that is modeled by this system. I start from the step response of this critically damped system:

$$\delta(t) = \frac{1}{K_{steer}} \left[ 1 - \left( 1 + \frac{t}{\tau} \right) e^{-t/\tau} \right]$$

Using Eq 6, I can replace $K_{steer}$ by $I_{steer}/\tau^2$, such that I obtain

$$\delta(t) = \frac{\tau^2}{I_{steer}} \left[ 1 - \left( 1 + \frac{t}{\tau} \right) e^{-t/\tau} \right]$$

Our objective is to find the time at which the angular rate $\dot{\delta}(t)$ is the highest. This angular rate is the following:

$$\dot{\delta}(t) = \frac{\partial \delta}{\partial t} = \frac{t e^{-t/\tau}}{I_{steer}} \propto t e^{-t/\tau}$$

The strictly monotone transformation $ln\left( \dot{\delta}(t) \right)$ is a concave function of $t$, and therefore the maximum of $\dot{\delta}(t)$ can be found by solving

$$\frac{\partial ln\left( \dot{\delta}(t) \right)}{\partial t} = 0$$

The result of this equation is $t = \tau$. Thus, the time constant $\tau$ is the time after movement onset at which the speed is the highest.

### What are realistic turn radiuses, lean angles and steering angular rates?

I start from the following requirements: (1) the turn radius may not be less than the minimum radius below which the tires loose traction, (2) the lean angle may not exceed an upper bound above which most riders would feel uncomfortable, and (3) the steering angular rates may not exceed the ones that the fastest human hands can make. Starting with the first requirement, skidding occurs when the centrifugal force $mv^2/R$ exceeds the frictional force $m\mu g$, with $m$ being the mass of the bicycle-rider combination and $\mu$ the coefficient of friction. For rolling rubber tires on asphalt, $\mu = 0.75$ is a good choice [35], and this corresponds to a minimum turn radius of $v^2/\mu g = 4.3056^2/(0.75 \times 9.81) = 2.5196$ m. In the following, we will report the curvature, which is the inverse of the radius. The maximum curvature is $1/2.5196 = 0.3969\ m^{-1}$.

The curvature $C$ follows from the kinematics of the bicycle model (see *The kinematic model*). I use an equation that takes into account the location of the combined CoG along the longitudinal axis [30] and the fact that curvature depends on the steering axis angle $\lambda$ [36]:

$$C \approx \frac{\tan(\delta) \cos(\beta(\delta))}{W} h(\lambda, \phi_1)$$

in which $\beta(\delta)$ is the slip angle, which accounts for the location of the combined CoG along the longitudinal axis, and $h(\lambda, \phi_1)$ is a correction factor for a non-vertical steering axis ($\lambda \neq \pi/2$). In the SDP, $\lambda = \pi/2$ and $h(\lambda, \phi_1) = 1$; in the BDP, $\lambda \neq \pi/2$ and $h(\lambda, \phi_1) = \cos(\pi/2 - \lambda)/\cos(\phi_1)$.

I now determine an upper bound for the combined CoG lean angle above which most riders would feel uncomfortable. Based on informal observations, I start from a rider that makes a steady U-turn at 15.5 km/h (= 4.3056 m/s) on a 7 m. wide two-way road, which is a regular width in Europe. To stay balanced in such a turn, the combined CoG lean angle must produce a gravitational acceleration that balances the turn-induced centrifugal acceleration. A simple geometrical argument shows that this lean angle is the following: $\tan^{-1}((v^2/R)/g) = \tan^{-1}((4.3056^2/7)/9.81) = 0.2637$ rad. (= 15.1072 degrees). Because the objective of the model is to keep the bicycle upright, and not to keep it in a steady U-turn, the average combined CoG lean angle must be substantially less than 0.2637 rad.

Finally, to find an upper limit for the steering angular rate, I start from the fastest hand movement observed in a reaching task, which is 4 m/s [37]. Combining this linear velocity with a typical commuter handlebar width of 0.6 m, I find a critical steering angular rate of 13.33 rad/s.

## Simulating the stabilization of the mechanical by the computational system

I have written computer code in Matlab for simulating the stabilization of the mechanical by the computational system and visualizing the results. This code is added to the supplementary information for this paper. With this code, one can perform all the simulations on which I have reported in this paper as well as variations inspired by one's own questions and hypotheses. Running simulations is only possible in discrete time, and I must therefore discretize the continuous time model. This is the main topic of this section.

**Simulating the combined system in discrete time.** The discrete time axis is defined by the increment $\Delta t$: 0, $\Delta t$, $2\Delta t$, $3\Delta t$ .... The model in Fig 5 involves a closed loop, and to describe it, one can start at every point. Here, I start from the sensory input system, which receives the state $x(t)$ from the mechanical system and feeds the noise-corrupted sensory input $y(t) = Cx(t) + s(t)$ into the computational system. This is depicted schematically in Fig 4. The computational system determines the internal state estimate $\hat{x}(t + \Delta t)$ on the basis of $y(t)$, the previous internal state estimate $\hat{x}(t - \Delta t)$, and the previous control action $u(t - \Delta t)$. No internal state estimate is calculated for time $t$. The new control action $u(t + \Delta t)$ is obtained from $\hat{x}(t + \Delta t)$.

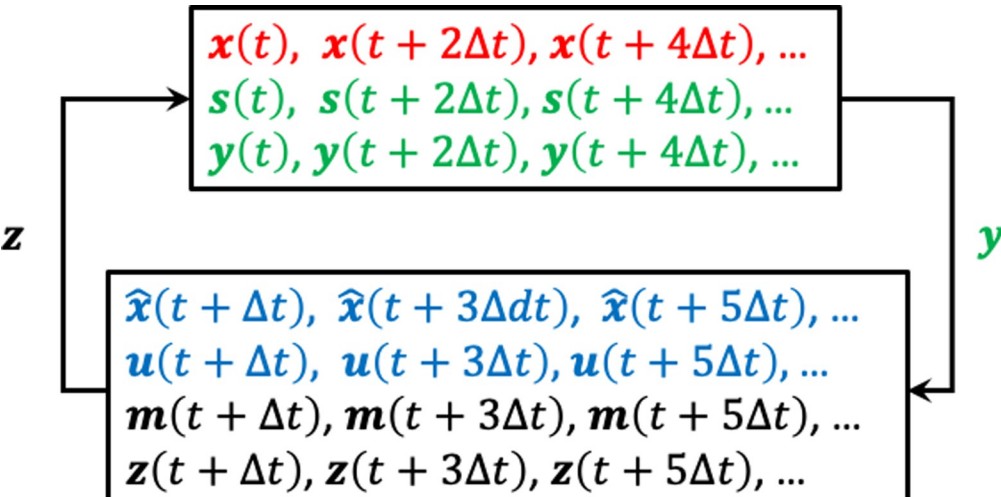

**Fig 4. Schematic representation of the simulation of the combined system in discrete time.** In red, green, blue and black, I show the variables that generated in, respectively, the mechanical, the sensory input, the computational, and the motor output system.

Adding the motor noise $m(t + \Delta t)$ to $u(t + \Delta t)$ produces $z(t + \Delta t)$, the input to the mechanical system. This input $z(t + \Delta t)$, together with the previous state $x(t)$ produces the new state $x(t + 2\Delta t)$. From this new state and the sensor noise $s(t + 2\Delta t)$, the new sensory input $y(t + 2\Delta t)$ is obtained, which closes the loop. No actual state and sensory input is calculated at time $t + \Delta t$.

**Solving the discrete time computational and mechanical system.** For simulating the combined system, one must solve the discrete time mechanical and computational system. For the mechanical system, this involves finding $x(t + 2\Delta t)$ by numerically integrating $\dot{x} = \Omega(x, z) = \Omega(x, u + m)$ over the interval $[t, t + 2\Delta t]$ starting from the initial condition $x(t)$ and with external input $u = u(t + \Delta t)$ and $m = m(t + \Delta t)$. For this, I used the Matlab function ode45, which is based on an explicit Runge-Kutta (4,5) formula [38].

To solve the discrete time computational system, I follow a similar approach, but now take advantage of the fact that an explicit solution exists for linear systems. Using this explicit solution, I can write the discrete time version of the linear approximation as follows:

$$x(t + \Delta t) = A_{2\Delta t}x(t - \Delta t) + B_{2\Delta t}u(t - \Delta t) + \Sigma_{2\Delta t}^{1/2}n^{(1)} \qquad \text{Eq 8}$$

$$y(t) = C_{2\Delta t}x(t - \Delta t) + \Psi_{2\Delta t}^{1/2}n^{(2)} \qquad \text{Eq 9}$$

The simulated versions of the actual motor and sensor noise are, respectively, $\Sigma_{2\Delta}^{1/2}n^{(1)}$ and $\Psi_{2\Delta}^{1/2}n^{(2)}$, with $n^{(1)}$ and $n^{(2)}$ denoting independent normally distributed random variables with an identity covariance matrix. The noises $\Sigma_{2\Delta t}^{1/2}n^{(1)}$ and $\Psi_{2\Delta t}^{1/2}n^{(2)}$ thus have a normal distribution with respective covariance matrices $\Sigma_{2\Delta t}$ and $\Psi_{2\Delta t}$, which are defined as follows:

$$\Sigma_{2\Delta t} \approx \int_0^{2\Delta t} e^{A\tau}\Sigma e^{A^T\tau}d\tau$$

$$\Psi_{2\Delta t} = (2\Delta t)\Psi$$

The matrices $A_{2\Delta t}$, $B_{2\Delta t}$, and $C_{2\Delta t}$ follow from the well-known solution of a linear state-space model with defining matrices $A$, $B$, and $C$: $A_{2\Delta t} = e^{A(2\Delta t)}$, $B_{2\Delta t} = A^{-1}(A_{2\Delta t} - I)B$, and $C_{2\Delta t} = C$ [39]. Note that the sensory input $y$ (see Fig 4) is evaluated at a different time than the simulated state variable $\hat{x}$, because the former is obtained from the mechanical system.

In one of the simulation studies (*Is the model robust to inaccuracies in the learned noise covariance matrices $\Sigma$ and $\Psi$?*), I used the optimal learned motor and sensor noise covariance matrices $\Sigma_{2\Delta t}$ and $\Psi_{2\Delta t}$. For the continuous time case, these optimal learned noise covariance matrices are the following functions of the actual noise covariance matrices $\Phi$ and $\Xi$: $\Sigma = B\Phi B^T$ and $\Psi = \Xi$. For the discrete time case, the corresponding formulas are the following:

$$\Sigma_{2\Delta t} \approx \int_0^{2\Delta t} e^{A\tau}B\Phi B^T e^{A^T\tau}d\tau$$

$$\Psi_{2\Delta t} = (2\Delta t)\Xi$$

For the discrete time computational system in Eqs 8 and 9, I calculate control actions $\boldsymbol{u}$ that minimize a cost functional $J_{2\Delta t}$:

$$J_{2\Delta t} = \lim_{N \to \infty} \frac{1}{N} \mathcal{E} \left( \sum_{n=1}^{N} \left[ \boldsymbol{x}(n2\Delta t - \Delta t)' Q \boldsymbol{x}(n2\Delta t - \Delta t) + \boldsymbol{u}(n2\Delta t - \Delta t)' R \boldsymbol{u}(n2\Delta t - \Delta t) \right] \right)$$

The cost functional $J_{2\Delta t}$ is minimized by control actions $\boldsymbol{u} = -M_{2\Delta t}\hat{\boldsymbol{x}}$, in which $-M_{2\Delta t}$ is the discrete time LQR gain (which depends on the matrices $A_{2\Delta t}$, $B_{2\Delta t}$, $Q$, and $R$), and $\hat{\boldsymbol{x}}$ is the optimal state estimate defined by this discrete time ODE:

$$\hat{\boldsymbol{x}}(t + \Delta t) = (A_{2\Delta t} - B_{2\Delta t} M)\hat{\boldsymbol{x}}(t - \Delta t) + K_{2\Delta t}[\boldsymbol{y}(t) - C_{2\Delta t}\hat{\boldsymbol{x}}(t - \Delta t)]$$

The matrix $K_{2\Delta t}$ is the discrete time Kalman gain, which depends on $A_{2\Delta t}$, $C_{2\Delta t}$, $\Sigma_{2\Delta t}$, and $\Psi_{2\Delta t}$.

**Discrete time motor and sensor noise.** From the properties of a Wiener process, it is straightforward to obtain the discrete time motor and sensor noise from the continuous time equations Eqs 1 and 2:

$$\boldsymbol{z}(t + \Delta t) = \boldsymbol{u}(t + \Delta t) + \Phi_{2\Delta t}^{1/2}\boldsymbol{n}^{(1)}$$

$$\boldsymbol{y}(t) = C\boldsymbol{x}(t) + \Xi_{2\Delta t}^{1/2}\boldsymbol{n}^{(2)}$$

The noises $\Phi_{2\Delta t}^{1/2}\boldsymbol{n}^{(1)}$ and $\Xi_{2\Delta t}^{1/2}\boldsymbol{n}^{(2)}$ have a normal distribution with respective covariance matrices $\Phi_{2\Delta t} = (2\Delta t)\Phi$ and $\Xi_{2\Delta t} = (2\Delta t)\Xi$.

## Results

### Stabilizing a nonlinear mechanical system by linear stochastic OFC

**A model for sensorimotor control.** The EoM for most dynamical systems are nonlinear. This holds for the SDP model bicycle, but also for common movements such as reaching, throwing, and walking; these movements are all performed by changing joint angles, which results in EoM involving trigonometric functions. I denote the nonlinear EoM as follows:

$$\dot{\boldsymbol{x}} = \Omega(\boldsymbol{x}, \boldsymbol{u})$$

The vector $\boldsymbol{x}$ contains the state variables, and $\dot{\boldsymbol{x}}$ their first derivatives with respect to time. For the SDP, $\boldsymbol{x} = \left[\delta, \phi_1, \phi_2, \dot{\delta}, \dot{\phi_1}, \dot{\phi_2}\right]^T$ and $\boldsymbol{u} = \left[T_\delta, T_{\phi_2}\right]^T$ (see Fig 2). In the Methods and Models section, I derive the SDP EoM from Lagrangian mechanics.

OFC calculates optimal control actions $\boldsymbol{u}$ that minimize a cost functional $\boldsymbol{J}(\boldsymbol{x}(\cdot), \boldsymbol{u}(\cdot))$, in which $\boldsymbol{x}(\cdot)$ and $\boldsymbol{u}(\cdot)$ denote the trajectories of, respectively, the state variables and the control actions. Typically, this cost functional increases with the integrated imprecision and energetic costs (e.g., the integrated squared length of $\boldsymbol{x}(\cdot)$, resp., $\boldsymbol{u}(\cdot)$; see further). Crucially, this cost functional depends on the EoM, and this raises the important question how the CNS can calculate optimal control actions in the extremely likely scenario that it does not know $\Omega(\boldsymbol{x}, \boldsymbol{u})$ exactly. For this scenario, I assume that the CNS learns an approximation to $\Omega(\boldsymbol{x}, \boldsymbol{u})$ from experience with the mechanical system. The CNS then uses this approximation as an internal model to estimate the state and calculate the optimal control actions.

In Fig 5, I have depicted a model for sensorimotor control that is based on an internal model that is a linear approximation of the unknown nonlinear dynamics $\Omega(\boldsymbol{x}, \boldsymbol{z})$. These nonlinear dynamics are depicted in red and will be denoted as the mechanical system. In its

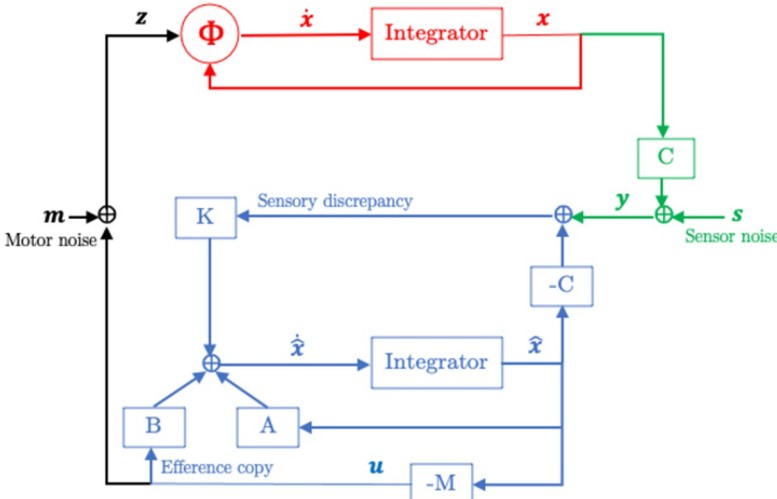

**Fig 5. Sensorimotor control of a mechanical system (in red) by input from a computational system (in blue).** The mechanical system is governed by the nonlinear differential equations $\dot{x} = \Omega(x, z)$, and the computational system produces an optimal control action $u$. The motor output system (in black) adds noise $m$ to $u$ and feeds this into the mechanical system. The sensory input system (in green) maps the state variables $x$ to sensory variables, adds noise $s$ and feeds the resulting sensory input $y$ into the computational system. The computational system calculates an optimal internal state estimate $\hat{x}$ by integrating a linear differential equation (characterized by the matrices $A$, $B$, $C$, and the Kalman gain $K$) that takes the sensory feedback $y$ as input. The optimal control action $u$ is obtained from $\hat{x}$ and the linear quadratic regulator (LQR) gain $-M$.

application to balancing a bicycle, this mechanical system corresponds to the rider's body plus his bicycle, and the linearization is around the equilibrium point (unstable fixed point). In other applications, the mechanical system may also involve objects in the environment that are sensed from a distance using vision and/or audition, and the linearization may be around a trajectory. The mechanical system receives input $z$ from the motor output system (in black), which adds noise $m$ to the optimal control action $u$. The sensory input system (in green) maps the state variables $x$ onto sensory variables (as specified by the matrix $C$), adds noise $s$ and feeds the resulting sensory input $y$ into the computational system (in blue).

The computational system consists of two components: (1) the internal model, which calculates an optimal internal state estimate $\hat{x}$ by integrating a linear differential equation (characterized by the matrices $A$, $B$, $C$ and the Kalman gain $K$) that takes the sensory feedback $y$ as input, and (2) the feedback control law, which determines the control action $u$ by multiplying the state estimate $\hat{x}$ with the LQR gain $-M$ (minus sign added for consistency with the existing literature). The matrices $A$, $B$ and $C$ must be learned from experience with the mechanical system. Useful reference values for $A$ and $B$ can be obtained from the first order Taylor approximation of the nonlinear $\Omega(x, u)$ at the unstable fixed point $x = 0$ and $u = 0$. That is, $\Omega(x, u)$ can be linearly approximated by $Ax + Bu$, with $A$ and $B$ being the Jacobian of $\Phi(x, u)$ at the unstable fixed point.

**Motor and sensor noise.** The stabilizing performance of the combined mechanical-computational system (i.e., how close $x$ stays to its target value) is adversely affected by motor and sensor noise: motor noise directly feeds into the mechanical system, and sensor noise degrades the internal state estimate. The model for the motor input $z$ to the mechanical system is a simple errors-in-variables model: $z = u + m$. And the model for the sensory input $y$ to the computational system is the linear model $y = Cx + s$. All variables are functions of continuous time. I assume that the noise terms $m$ and $s$ are linear combinations of independent vector-

valued Wiener processes $\boldsymbol{v}^{(1)}$ and $\boldsymbol{v}^{(2)}$:

$$z = \boldsymbol{u} + \Phi^{1/2} d\boldsymbol{v}^{(1)} \qquad \text{Eq 10}$$

$$\boldsymbol{y} = C\boldsymbol{x} + \Xi^{1/2} d\boldsymbol{v}^{(2)} \qquad \text{Eq 11}$$

The scaling matrices $\Phi^{1/2}$ and $\Xi^{1/2}$ determine the covariance of the motor noise $\boldsymbol{m} = \Phi^{1/2} d\boldsymbol{v}^{(1)}$ and the sensor noise $\boldsymbol{s} = \Xi^{1/2} d\boldsymbol{v}^{(2)}$. Specifically, the motor and the sensor noise are normally distributed with covariance matrices $\Phi dt$ and $\Xi dt$, respectively.

**Stochastic OFC deals with noise in an optimal way.** Stochastic OFC provides the tools to deal with motor and sensor noise, and it does so in an optimal way if the noise is Gaussian and additive [21]. This optimality is central to the proposed model for sensorimotor control, which I now formulate with the detail that is required to simulate it on a computer:

1. The CNS learns from experience the following matrices: $A$, $B$, $C$, and the covariances of the motor and the sensor noise. For the purposes of this paper, the matrix $C$ that maps $\boldsymbol{x}$ onto $\boldsymbol{y}$ is assumed to be known. The learned noise covariance matrices can be given plausible values, as I will describe in the Results section.

2. The control actions are produced by an internal model that is based on the following linear approximation of the other three systems:

$$\dot{\boldsymbol{x}} = A\boldsymbol{x} + B\boldsymbol{u} + \Sigma^{1/2} d\boldsymbol{w}^{(1)} \qquad \text{Eq 12}$$

$$\boldsymbol{y} = C\boldsymbol{x} + \Psi^{1/2} d\boldsymbol{w}^{(2)} \qquad \text{Eq 13}$$

in which $\boldsymbol{w}^{(1)}$ and $\boldsymbol{w}^{(2)}$ are independent vector-valued Wiener processes. The terms $\Sigma^{1/2} d\boldsymbol{w}^{(1)}$ and $\Psi^{1/2} d\boldsymbol{w}^{(1)}$ are simulated versions of the motor and sensor noise. These noise terms are normally distributed with covariance matrices $\Sigma dt$ and $\Psi dt$, respectively. The matrix $\Sigma$ represents the learned amplitude of the movement inaccuracies that are produced by a noisy motor input ($\boldsymbol{u}$ + noise), and the matrix $\Psi$ represents the learned amplitude of the sensory discrepancies ($\boldsymbol{y}$–$C\boldsymbol{x}$). Although the actual and the simulated state may differ, I will use the same state variable $\boldsymbol{x}$ for the mechanical model $\dot{\boldsymbol{x}} = \Omega(\boldsymbol{x}, \boldsymbol{z})$ as for the linear model in Eqs 12 and 13. The representation of this linear model in Eqs 12 and 13 is called a state-space representation.

3. The CNS calculates the control action $\boldsymbol{u}$ such that a cost functional $J$ is minimized:

$$J = \lim_{T \to \infty} \frac{1}{T} \mathcal{E} \left( \int_0^T \left[ \boldsymbol{x}(t)' Q\boldsymbol{x}(t) + \boldsymbol{u}(t)' R\boldsymbol{u}(t) \right] dt \right) \qquad \text{Eq 14}$$

in which $\varepsilon$ () denotes expected value, and $Q$ and $R$ are positive definite matrices of the appropriate dimensions. The component $\boldsymbol{x}(t)' Q\boldsymbol{x}(t)$ quantifies the precision of the internal state variable $\boldsymbol{x}$ when the target state equals $\boldsymbol{0}$; for the general case of a target state equal to $\boldsymbol{c}$, this component is $[\boldsymbol{x}(t)–\boldsymbol{c}]' Q[\boldsymbol{x}(t)–\boldsymbol{c}]$. The component $\boldsymbol{u}(t)' R\boldsymbol{u}(t)$ quantifies the energetic cost.

4. Under the linear model in Eqs 12 and 13, the cost functional $J$ is minimized by control action $\boldsymbol{u} = -M\hat{\boldsymbol{x}}$, in which $-M$ is the LQR gain, and $\hat{\boldsymbol{x}}$ is an optimal state estimate defined

by this ODE:

$$\dot{\hat{x}} = (A - BM)\hat{x} + K(\boldsymbol{y} - C\hat{x})$$

The term $(A - BM)\hat{x} = A\hat{x} + B\boldsymbol{u}$ only depends on the internal model, and the term $K(\boldsymbol{y} - C\hat{x})$ also depends on the sensory feedback $\boldsymbol{y}$. The matrix $K$ is the Kalman gain, which depends on $A$, $C$, $\Sigma$, and $\Psi$, the covariance matrices of the learned motor and sensor noise. The LQR gain $-M$ depends on the matrices $A$, $B$, $Q$, and $R$.

Motor and sensor noise have both a direct and an indirect effect on the stabilizing performance of the combined system: (1) motor and sensor noise directly feed into, respectively, the mechanical and the computational system, and (2) via the Kalman gain $K$, the state estimate $\hat{x}$ depends on the internal covariance matrices $\Sigma$ and $\Psi$, which the CNS must learn from experience with the actual motor and sensor noise. The importance of this learning process follows from the fact that the accuracy of $\Sigma$ and $\Psi$ has a positive effect on the stabilizing performance of the computational system. This fact can be proved for a linear mechanical system, and it is approximately true for a nonlinear mechanical system in a region of the state-space for which this system is approximately linear. Specifically, for a linear mechanical system, $\Phi(\boldsymbol{x}, \boldsymbol{z}) = A\boldsymbol{x} + B\boldsymbol{z}$, and using Eq 10, this system can be rewritten as $\Phi(\boldsymbol{x}, \boldsymbol{z}) = A\boldsymbol{x} + B\boldsymbol{u} + B\,\Phi^{1/2}\,d\boldsymbol{v}^{(1)}$. Comparing this with the first state-space equation of the computational system (Eq 12), we see that the two systems are identical if $\Sigma = B\Phi B^{\mathrm{T}}$. In addition, comparing Eq 11 and Eq 13, we see that the sensory system is identical to the state-space equation of the computational system if $\Psi = \Xi$. Thus, optimal control of a linear mechanical system involves a Kalman gain that is calculated using $\Sigma = B\Phi B^{T}$ and $\Psi = \Xi$.

**Is the optimal model good enough?.** For stochastic OFC to be a good model for bicycle balance control, the bicycle and the rider must remain balanced over a range of lean and steering angles that is observed. Importantly, the optimality of stochastic OFC does not automatically ensure that the model is also good enough in that respect [40]. The performance of the model depends on how well the linear internal model approximates the external nonlinear dynamical system plus the motor and the sensor noise covariance matrices. The accuracy of the approximation in turn depends on two factors: (1) how good is the linear approximation with optimal values for the linear model's parameters $A$, $B$, $C$, $\Sigma$ and $\Psi$, and (2) how close are the actual values to these optimal parameter values? The performance of the optimal linear approximation is investigated in the first of three simulation studies. Specifically, in this simulation study, I will evaluate whether stochastic OFC with optimal parameter values can balance the model bicycle for steering and lean angles that are observed with real riders on real bicycles, without requiring steering angular rates that no real rider can produce. However, it is unlikely that the linear model's parameters are exactly at their optimal values, and the possible consequences of this are discussed next.

**Which parameters are responsible for stabilization failures?.** Stabilization may fail (i.e., bicycle and rider fall over) because of motor and sensor noise. However, stabilization also depends on the parameters of the computational system, and in this paper, I will investigate the role of a few of these parameters. The computational system is fully specified by the following seven matrices: $A$, $B$, $C$, $\Sigma$, $\Psi$, $Q$ and $R$, and I will investigate the role of the following three: $A$, $\Sigma$ and $\Psi$.

It is useful to distinguish between the static and the dynamic parameters of the computational system: the dynamic parameters are the state variables $\boldsymbol{x}$, and the static parameters are the seven matrices on which these state variables depend. From a theoretical perspective, it is a matter of choice whether a parameter is considered static or dynamic. However, from an

applied perspective (here, bicycle balance control), it is important to know the time scale over which the parameters are likely to change. This is related to the robustness of the computational system: if the system is not robust to inaccuracies in some dynamic parameter, then the organism needs a mechanism to correct these inaccuracies. If this mechanism is slow (more than a day), it is usually called "learning" (offline updating), and if it is fast, it is usually called "sensory feedback" (online updating). For the model considered here, the CNS must learn the internal noise covariance matrices $\Sigma$ and $\Psi$ from experience with the mechanical system and the motor and sensor noise. The required learning rate is set by the robustness of the computational system: the more robust the computational system to inaccuracies in $\Sigma$ and $\Psi$, the slower the learning rate may be.

The matrix $A$ depends on the bicycle speed $v$, which I assume to be constant when calculating the Kalman and the LQR gain. However, in reality, $A$ is a dynamic parameter because the bicycle speed $v$ changes over time: $A = A(v) = A(v(t))$. The time scale of the changes in bicycle speed is in the order seconds (e.g., accelerating from 0 to 1.5 m/sec. takes about 1 sec.). Thus, the CNS probably needs online updates of the bicycle speed. Crucially, the robustness of the computational system to inaccuracies in the speed estimates becomes more important as these updates are less reliable. This is relevant here, because there is good psychophysical evidence against reliable speed estimates based on optical flow [9]. Thus, a plausible computational system must be robust to inaccurate speed estimates.

In sum, the CNS must learn and/or estimate some parameters of the computational system. Because this process takes time, the system's performance must be robust to inaccuracies in the system's parameters. I investigated this robustness in two simulation studies in which I manipulated the accuracy of (1) the learned noise covariance matrices $\Sigma$ and $\Psi$, and (2) the system (state) matrix $A$. These two parameter sets correspond to two different aspects of the environment that the CNS must learn: (1) the reliability of the motor output and the sensory input, and (2) the physical laws that govern the movements of our body and bicycle. They also play different roles in the computational model: the learned noise covariance matrices only affect the Kalman gain (which updates the internal state estimate), whereas the learned system matrix also affects the LQR gain (which maps the state estimate on the control action).

## How plausible is stochastic OFC as a model for sensorimotor control?

To evaluate the plausibility of stochastic OFC as a model for sensorimotor control, in three simulation studies, I address the following questions: (1) Is the optimal model good enough to balance a bicycle under realistic conditions, and (2) Is the model robust against inaccuracies in the model parameters? I begin by describing what I mean by "realistic conditions".

**What are realistic turn radiuses, lean angles, and steering angular rates?.** For our model to be plausible, it must balance the model bicycle for lean angles that approach the values observed with real riders on real bicycles, without requiring turn curvatures (inverse turn radiuses) and steering angular rates that cannot be produced. To determine critical values for these parameters, I put forward the following requirements: (1) the turn curvatures may not exceed the maximum curvature above which the tires loose traction (i.e., skid), (2) the lean angle may not exceed an upper bound above which most riders would feel uncomfortable, and (3) the steering angular rates may not exceed the ones that the fastest human hands can produce.

In the *Materials and Methods*, I give a quantitative rationale for the maximum curvature, the maximum combined CoG lean angle, and maximum angular rate: 0.3969 m$^{-1}$, 0.2637 rad. and 13.33 rad/s, respectively. In all simulations, the trials without skidding (i.e., curvature everywhere less than 0.3969 m$^{-1}$) had steering angular rates that were more than an order of

magnitude smaller than the critical steering angular rate 13.33 rad/s. More detailed results will therefore only be shown for the curvatures and the combined CoG lean angles.

**What are plausible parameter values for the OFC cost functional?.** I calculated the LQR gain for a cost functional that implements the objective that the combined CoG must be kept over the LoS. Because the LQR cost functional is a quadratic form, an objective with respect to the combined CoG lean angle must be expressed as a linear function of the state variables. Because the combined CoG lean angle is a nonlinear function of the state variables $\phi_1$ and $\phi_2$, I approximated it by a linear Taylor series approximation in which I inserted the lengths and masses used in the simulations. The following linear approximation resulted: $0.821 \times \phi_1 + 0.179 \times \phi_2$.

I used a block-diagonal precision matrix $Q$ (see Eq 14) with the following submatrix for the angles $[\delta, \phi_1, \phi_2]$:

$$\mathrm{diag}\left( w_\delta, \begin{bmatrix} 0.821^2 & 0.821 \times 0.179 \\ 0.179 \times 0.821 & 0.179^2 + w_{\phi_2} \end{bmatrix} \right)$$

The weights $w_\delta$ and $w_{\phi_2}$ quantify the importance of keeping $\delta$ and $\phi_2$ close to 0 relative to the importance of keeping the combined CoG close to 0. Because the steering angle is not involved in the balancing objective, I chose a very small value for $w_\delta$: $w_\delta = 0.001$. I chose the value 1 for $w_{\phi_2}$, which assigns an equal importance to the balance objective with respect to the combined CoG, and the one with respect to $\phi_2$ (see *Cycling involves a double balance problem*). Drastically increasing $w_{\phi_2}$ (to $w_{\phi_2} = 100$) improved the stabilization of both $\phi_2$ and the combined CoG lean angle (keeping them closer to 0), as quantified by the stabilization metrics of the simulation study (see further). Because the focus of the present paper is on the robustness of control based on an internal model, this effect will not be investigated any further. Finally, I used the same submatrix for the angular rates $\left[\dot{\delta}, \dot{\phi}_1, \dot{\phi}_2\right]$ as for the corresponding angles $[\delta, \phi_1, \phi_2]$, which implements the objective that it is equally important to keep the angles stationary as it is to keep them close to 0.

The LQR gain also depends on the matrix $R$, which quantifies the relative importance of the energetic cost (see Eq 14). I ran my simulations with $R = \mathrm{diag}([1, 1])$. Increasing the diagonal elements of $R$ reduces the stabilization performance.

**Is the optimal model good enough to balance a bicycle under realistic conditions?.** To evaluate the plausibility of the model, I simulated state variables for increasing noise amplitudes, which produced increasing lean and steering angles. I evaluated whether, over the increasing noise amplitudes, the average combined CoG lean angle remains well below angles at which most riders start feeling uncomfortable (0.2637 rad.) without skidding (i.e., curvatures exceeding 0.3969 m$^{-1}$).

Noise enters the mechanical system via the motor output $z$ and the sensory input $y$, and its amplitude is determined by the motor and the sensor noise covariance matrices $\Phi$ and $\Xi$. The dimensions of $\Phi$ correspond to the two control actions, steering and upper body lean torque ($\boldsymbol{u} = \left[T_\delta, T_{\phi_2}\right]^T$), and the dimensions of $\Psi$ correspond to the six sensory inputs. I independently varied the amplitudes of three different noise types: steering noise, upper body noise, and sensor noise. I did this by specifying $\Phi$ and $\Psi$ as diagonal matrices defined by three scalar constants, $c_\delta$, $c_{\phi_2}$ and $c_y$ : $\Phi = \mathrm{diag}\left(\left[c_\delta, c_{\phi_2}\right]\right)$, and $\Psi = \mathrm{diag}([c_y, c_y, c_y, c_y, c_y, c_y])$. There were only small differences between the three noise types with respect to how much they affected the lean and the steering angles. These differences did not justify a discussion of the more

complicated pattern of results as compared to the results for homogeneous noise amplitudes, $c_\delta = c_{\phi_2} = c_y = c$.

I evaluated the plausibility of the model at its optimal parameter values. Specifically, the matrices $A$ and $B$ were set equal to the Jacobian of the EoM at the unstable fixed point, and the learned motor and sensor noise covariance matrices $\Sigma$ and $\Psi$ were given values that correspond to the actual motor and sensor noise covariance matrices $\Phi$ and $\Xi$ (see *Which learned parameters are responsible for stabilization failures?*).

I linearly increased the values of the noise amplitude $c$ from 0.001 to 0.05 and simulated the model under the resulting motor and sensor noise. For every noise amplitude, I simulated 100 trials of 60 seconds at $\Delta t = 0.01$. As expected, with increasing noise amplitude, also the number of trials with skidding increased (see Fig 6A). The rest of the results is based on the successful (no skidding) trials, for which I quantified the model's performance by the root-mean-square (RMS) combined CoG lean angle and the maximum curvature. These numbers were subsequently averaged over the trials. As expected, both the RMS combined CoG lean angle and the maximum curvature increased with the noise amplitude (see Fig 6C). Crucially, even for the highest noise level, the RMS combined CoG lean angle was well below its upper bound (0.2637 rad.).

I next investigated whether the results for the SDP generalize to a linearized model with self-stabilizing forces, the BDP. The BDP is a combination of an existing benchmark model for studying the passive dynamics of a bicycle [11] and the double pendulum.

The simulations for the BDP were performed in the same way as for the SDP, and the results are shown in Fig 6B and 6D. Crucially, to obtain approximately the same percentage of skid trails in the BDP as in the SDP simulations, the noise amplitude had to be increased by a factor of approximately 14 (compare the x-axes of Fig 6A and 6B). This shows that the BDP is much less susceptible to noise than the SDP. This is most likely due to the positive trail of the BDP, which is responsible for caster forces in the front frame. Caster forces reduce the impact of the noise on the handlebars because they align the front wheel with the rear frame [36].

Except for the reduced susceptibility to noise, the results for the BDP are like those for the SDP: even for the highest noise level, the RMS combined CoG lean angle is well below the

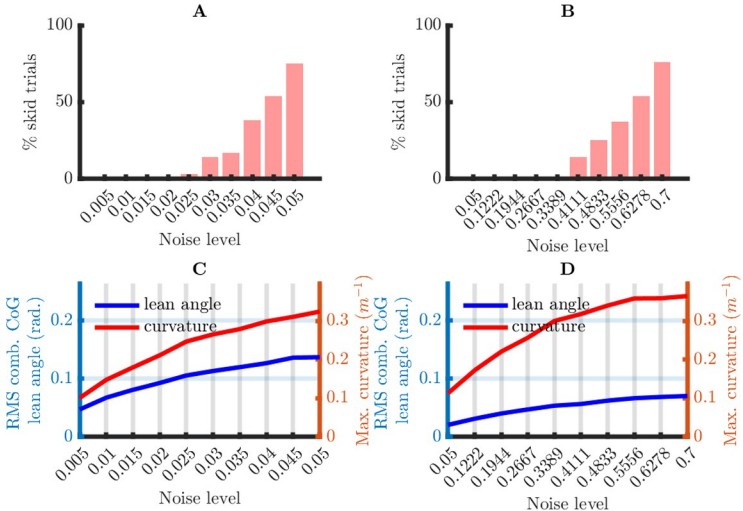

**Fig 6. Simulation results for the model at its optimal parameter values.** (A, B) Percentage of trials with skidding, separately for the SDP (in A) and BDP (in B) simulations. (C, D) RMS combined CoG lean angle and maximum curvature, averaged over the successful trials in the SDP (in C) and BDP (in D) simulations.

upper bound for a comfortable lean angle (0.2637 rad.), and even lower than for the SDP. Thus, stochastic OFC with optimal parameter values can balance a bicycle under realistic conditions and this does not depend on the bicycle model.

**Is the model robust to inaccuracies in the learned noise covariance matrices?.** I investigated the robustness to inaccuracies in $\Sigma$ and $\Psi$ by systematically varying the difference between these parameters and their corresponding optimal values $\Sigma = B\Phi B^T$, $\Psi = \Xi$. I ran the study with actual motor and sensor noise covariance matrices $\Phi = \mathrm{diag}\left(\left[c_\delta, c_{\phi_2}\right]\right)$ and $\Xi = \mathrm{diag}([c_y, c_y, c_y, c_y, c_y, c_y])$. For the SDP simulations, I set $c_\delta = c_{\phi_2} = c_y = 0.035$. I manipulated the accuracy of $\Sigma$ and $\Psi$ by means of a noise fraction $f$ with logarithmically spaced values between 0.1 and 10 (two orders of magnitude). I investigated two types of inaccuracy: motor noise inaccuracy ($\Sigma = fB\Phi B^T$) and sensor noise inaccuracy ($\Psi = f\Xi$).

The results for the SDP are shown in Fig 7, separately for the manipulations of the learned motor noise $\Sigma$ (panels A and C) and those of the learned sensor noise $\Psi$ (panels B and D). For both noise types, the model performed best when the learned and the actual motor noise were equal. This effect on performance is only visible in the percentage of skid trials; the RMS combined CoG lean angle remained well below its upper bound (0.2637 rad.). Interestingly, there was an asymmetry between the motor and the sensor noise in the model's performance as a function of the learned noise fraction: suboptimal learned motor noise reduced performance much less when it was too small whereas suboptimal learned sensor noise reduced performance much less when it was too large. Thus, model-based balance control for the SDP has a specific type of robustness to inaccuracies in the learned noise covariance matrices: the stabilization is robust to learned motor noise covariances that are too small and learned sensor noise covariances that are too large.

I next investigated whether the results for the SDP generalize to the BDP. For these simulations, I set $c_\delta = c_{\phi_2} = c_y = 0.4833$. The results are shown in Fig 8. For the BDP, the model's performance was unaffected by the difference between the learned and the actual noise.

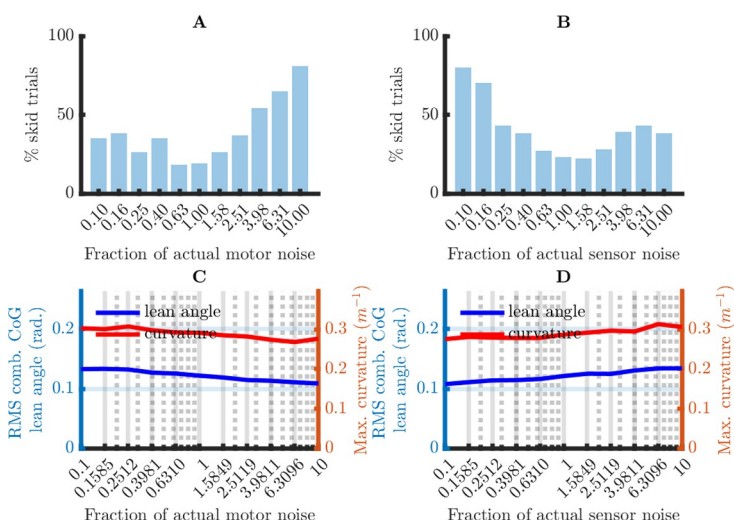

**Fig 7. Simulation results for the SDP model with learned noise covariance matrices at 11 logarithmically spaced fractions of the actual noise covariance matrices.** (A, B) Percentage of trials in which skidding occurred, separately for trials in which the learned motor noise $\Sigma$ (in A) and the learned sensor noise $\Psi$ (in B) was manipulated. (C, D) RMS combined CoG lean angle and maximum curvature, averaged over the successful trials in which the learned motor noise (in C) and the learned sensor noise (in D) was manipulated.

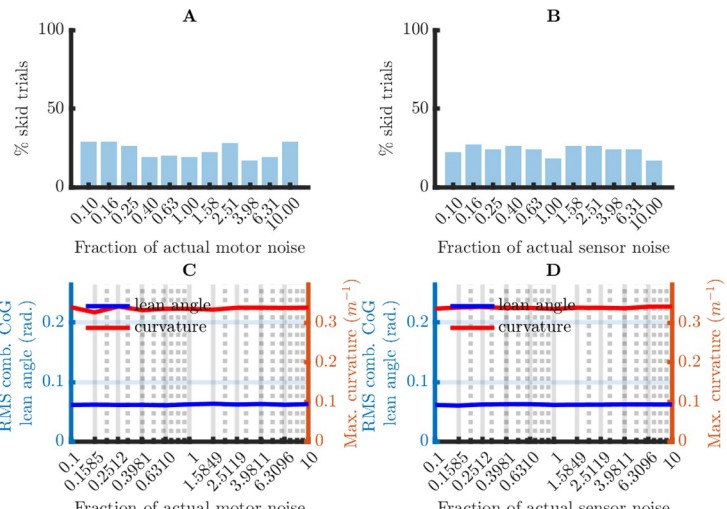

**Fig 8. Simulation results for the BDP model with learned noise covariance matrices at 11 logarithmically spaced fractions of the actual noise covariance matrices.** See the caption of Fig 7.

In sum, both for the SDP and the BDP, the stabilization is robust to inaccuracies over two orders of magnitude for the learned motor and sensor noise covariances. For the SDP, the stabilization is only robust to learned motor noise covariances that are too small and learned sensor noise covariances that are too large. For the BDP, this robustness is uniform.

**Is the model robust to inaccuracies in the system matrix due to speed misestimation?.** To investigate the robustness to inaccuracies in the system matrix $A$ I used the fact that the optimal system matrix (the Jacobian of $\Omega(\boldsymbol{x}, \boldsymbol{u})$ with respect to $\boldsymbol{x}$ and evaluated at the unstable fixed point) depends on the bicycle speed $v$ via the centrifugal acceleration (Eqs 1 and 3). I simulated a SDP with an actual speed of $v = 4.3$ m/sec., and calculated 13 different inaccurate system matrices $A$ by evaluating the Jacobian of $\Omega(\boldsymbol{x}, \boldsymbol{u})$ at linearly spaced values of $v$ between 90 and 110 percent of the actual speed. For the SDP simulations, I set $c_\delta = c_{\phi_2} = c_y = 0.015$, for which no skidding occurs when the optimal system matrix is used (see Fig 6).

The results in Fig 9 (panels A and C) show that SDP stabilization strongly depends on the accuracy of the speed estimate: successful trials were only found for speed fractions between 0.917 (51% completed) and 1 (100% completed). The robustness is asymmetrical around the true speed: there is a small range of underestimated speeds (fractions 0.9333 to 1) that allow for stabilization, but for overestimated speeds this range is much smaller (less than from 1 to 1.0167).

For the BDP simulations, I set $c_\delta = c_{\phi_2} = c_y = 0.1944$, for which no skidding occurs when the optimal system matrix is used. The pattern of results for the BDP is like the one for the SDP (see Fig 9, panels B and D) but the range of speed fractions that allows for BDP stabilization is much wider than for the SDP: from 0.725 to 1.1625. The risk for stabilization failures is again at the high end of the speed estimates.

In sum, compared to the robustness to inaccuracies in the learned noise covariance matrices (over two orders of magnitude), the stabilization is much less robust to inaccuracies in the system matrix that result from misestimation of the bicycle speed. This holds for both bicycle models.

## Discussion

I proposed and evaluated a model for sensorimotor control. The central concept in this model is a computational system, implemented in the CNS, that not only controls but also learns a

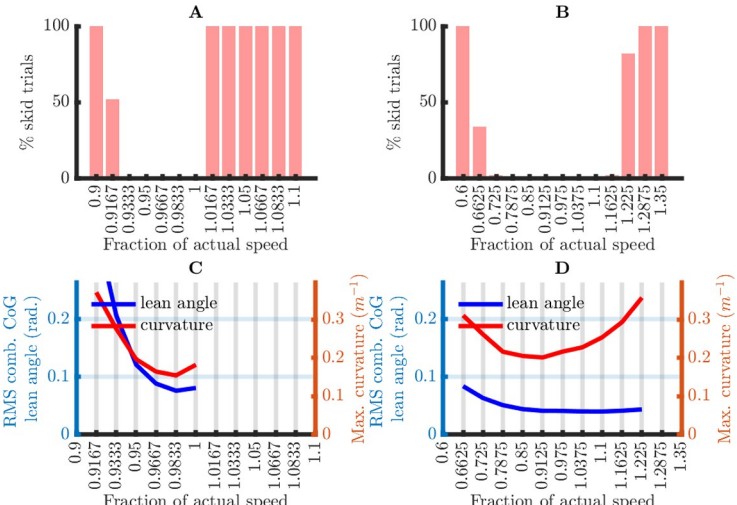

**Fig 9. Simulation results as a function of 11 linearly spaced fractions of the actual speed for which the system matrix *A* was calculated.** Across all simulations, the actual speed was kept constant at $v = 4.3$ m/sec. (A, B) Percentage of trials with skidding, separately for the SDP (in A) and BDP (in B) simulations. (C, D) RMS combined CoG lean angle and maximum curvature, averaged over the successful trials in the SDP (in C) and BDP (in D) simulations. Values are omitted for speed fractions at which no successful trials were obtained.

mechanical system that exists outside the CNS. At the interface between these two systems, there is a motor output system that transfers a control signal to the mechanical system, and a sensory system that maps the state of the mechanical system into the computational system. The computational system can simulate the combined mechanical, motor output, and sensory input system. It does so by means of a learned approximation of (1) the physical laws that govern the mechanical system, (2) the mapping performed by the sensory system, and (3) the reliability of the motor output and the sensory input. In my implementation of the computational system, I assumed that (1) the optimal learned approximation of the physical laws is linear, with the defining matrices (*A* and *B*) being the Jacobian of the EoM evaluated at the unstable fixed point, and (2) the optimal learned approximations of the internal noise covariance matrices are the noise covariance matrices of the optimal linear approximation of the combined system.

The control of the mechanical system by the computational system is optimal in the sense of stochastic OFC. It follows that the stabilization performance of the model only depends on three factors: (1) the amplitude of the motor output and the sensory input noise, (2) the optimality criterion (i.e., the expected cost-to-go), and (3) the accuracy of the learned approximation. Of these three, the accuracy of the learned approximation is the most interesting from a cognitive point of view, and the amplitude of the motor output and the sensory input noise is the most interesting from a physiological point of view.

I have applied this model to the balancing of a bicycle. This is not common in sensorimotor control, where the relevant data are often collected in experimental paradigms that ask for more isolated movements (e.g., reaching, pointing, lifting) that occur naturally as a part of more complex movements involving the whole body. Balancing a bicycle is more like walking but with the important advantage that the movements are strongly constrained by the geometry of the bicycle and the rider's position on it. It therefore does not come as a surprise that balancing a bicycle has become a topic of interest for non-academics with an interest in sensorimotor control; that community has contributed valuable observations by experimenting with the handling properties of a bicycle (e.g., by reversing the steering response).

Compared to the isolated movements in typical laboratory tasks, bicycle balance control has the additional advantage of societal relevance for the large group of senior citizens that want to maintain their mobility.

I conducted three simulation studies. In the first of these studies, I demonstrate that the model can balance a bicycle under realistic conditions. In the second study, I demonstrate that the model's stabilization performance is robust to inaccuracies in the learned noise covariance matrices. For the BDP, this robustness is uniform across the noise levels, but for the SDP, stabilization performance is only robust to learned motor noise covariances that are too small and learned sensor noise covariances that are too large. The third simulation study shows that, compared to the robustness to inaccuracies in the learned noise covariance matrices, the stabilization is much less robust to inaccuracies in the system matrix that result from misestimation of the bicycle speed.

All three simulation studies showed that the BDP is much less susceptible to noise than the SDP. This is probably due to the positive trail of the BDP, which generates caster forces in the front frame. These forces reduce the impact of the noise on the handlebars by aligning the disturbed front wheel with the non-disturbed rear frame [36].

As holds for every model, my model is only an approximation of reality. It is important to be aware of a few aspects for which I made a choice for the sake of computational feasibility or simplicity. Not all choices are inevitable, but more work is needed to extend the model, allowing it to perform all computations that are performed by the CNS. The first useful extension immediately follows from the third simulation study: if the model must apply to a wide range of speeds, a mechanism must be added for accurate speed estimation and selection of the appropriate system matrix. For motor control in general, this idea also has been proposed by Scott [41], but focusing only on the feedback (LQR) gain matrix (which depends on the system matrix). The question now is "What is the speed-related information based on which the appropriate system matrix is selected?". Because of its large Weber fraction [9], it is doubtful that optic flow is the only source of information for speed estimation. This is supported by the fact that many experienced cyclists can ride on stationary bicycle rollers. Moreover, they can do so while shifting gears, and this is inconsistent with a simple readout of proprioceptive information (pedaling cadence) as a substitute for the missing optic flow. I conclude that none of the known sources of sensory information is a plausible candidate for selecting the appropriate system matrix.

The second aspect to be aware of, is that the computational system is based on a linear approximation of the unknown mechanical system. Although it is difficult to argue against the idea that the internal model must be based on some sort of approximation, there is no reason that it should be linear and optimal for a single point (i.e., the unstable fixed point). For example, if it were the optimal linear approximation for the unstable fixed point, and the bicycle rider had learned the linear coefficients based on experience with lean angles below 5 degrees, then this linear approximation would also allow him to simulate the linear ODE in Eq 12 for much larger lean angles than he is familiar with. This would allow him to balance his bicycle outside the range he is familiar with. Whether this is possible, is still an empirical question.

The third aspect to be aware of pertains to the biological delays between the state estimate $\hat{x}$ and (1) the mechanical system input $z$ (the motor delay), and (2) the sensory feedback $y$ (the sensory delay). The motor delay is caused by the fact that the control action must pass via motor axons and muscles before it affects the mechanical system. And the sensory delay is caused by the fact that the sensory feedback must pass via a series of sensory neurons before it arrives in the computational system. In my model, I made the unrealistic assumption that both delays are zero. With respect to the motor delay, for a model that only estimates the current

state $\hat{\boldsymbol{x}}(t)$, the following must hold:

$$\boldsymbol{z}(t + T_{mot}) = -M\hat{\boldsymbol{x}}(t) + \boldsymbol{m}(t + T_{mot}),$$

in which $T_{mot}$ is the motor delay. Even for a small motor noise $\boldsymbol{m}$ and a state estimate $\hat{\boldsymbol{x}}$ that approximates the mechanical system states $\boldsymbol{x}$ very well, the torque $\boldsymbol{z}(t + T_{mot})$ will not stabilize the mechanical system if $\boldsymbol{x}(t + T_{mot})$ differs too much from $\boldsymbol{x}(t)$. This is a well-known problem in sensorimotor control, and it has been proposed that the prediction of future states may solve it [42–47]. This implies that the estimate $\hat{\boldsymbol{x}}(t)$ is replaced by a prediction $\tilde{\boldsymbol{x}}(t, T_{mot})$, which extrapolates the estimate at time $t$ (i.e., $\hat{\boldsymbol{x}}(t)$) to time $t + T_{mot}$.

With respect to the sensory delay, for a model that only estimates the current state $\hat{\boldsymbol{x}}(t)$, the following must hold:

$$\dot{\hat{\boldsymbol{x}}}(t) = (A - BM)\hat{\boldsymbol{x}}(t) + K[\boldsymbol{y}(t - T_{sens}) - C\hat{\boldsymbol{x}}(t)],$$

in which $T_{sens}$ is the sensory delay. Like the problem that is caused by a motor delay, if $\boldsymbol{x}(t-T_{sens})$ (the state reflected by $\boldsymbol{y}(t-T_{sens})$) differs too much from $\boldsymbol{x}(t)$, the state estimate $\hat{\boldsymbol{x}}(t)$ will be incorrectly updated. This problem can be solved by only updating the past state estimate $\hat{\boldsymbol{x}}(t - T_{sens})$. Combining this solution with the one for the motor delay, this results in a model in which the state estimate $\hat{\boldsymbol{x}}$ lags $T_{mot} + T_{sens}$ behind the true state $\boldsymbol{x}$, and the control action is calculated using the prediction $\tilde{\boldsymbol{x}}(t, T_{mot} + T_{sens})$. More work is required to evaluate whether the SDP and BDP can be balanced with a realistic motor and sensory delay, and whether prediction is necessary to achieve this.

The fourth aspect of the model to be aware of is that the control action is specified in torque values, whereas the output of the CNS are neuronal firing rates that are converted to joint torques by the muscles. This firing-rate-to-torque conversion is not a part of the model, and this most likely has consequences for the model's validity. For instance, in the computational model, the LQR gain performs a linear mapping from the state estimate to the control action, and this ignores the fact that the muscles may not be able to produce the required torques. This is especially important in the context of ageing and physical training, which affect the available torque ranges. Most likely, motor skill learning involves two parallel processes, one at the muscular level that determines the available torque ranges, and one at the level of the CNS that learns the mapping from the state estimate to the required torques. For the model to be valid, the CNS-level process must be informed by the available torque ranges.

It is possible to extend the model such that it incorporates the firing-rate-to-torque conversion, and this requires knowledge of the muscular physiology. Specifically, if the optimal control action $\boldsymbol{u}$ is a vector of firing rates, then one needs a new matrix $B$ that must be decomposable as follows:

$$B = B_{\dot{\boldsymbol{x}}\leftarrow\boldsymbol{u}} = B_{\dot{\boldsymbol{x}}\leftarrow\boldsymbol{\tau}}B_{\boldsymbol{\tau}\leftarrow\boldsymbol{u}},$$

in which $B_{\boldsymbol{\tau}\leftarrow\boldsymbol{u}}$ specifies the mapping from the firing rate vector $\boldsymbol{u}$ on the joint torques $\boldsymbol{\tau}$, and $B_{\dot{\boldsymbol{x}}\leftarrow\boldsymbol{\tau}}$ (the old matrix $B$) specifies the mapping from the joint torques on the state derivatives $\dot{\boldsymbol{x}}$. The matrix $B_{\boldsymbol{\tau}\leftarrow\boldsymbol{u}}$ must be specified based on knowledge of muscular physiology, and the matrix $B_{\dot{\boldsymbol{x}}\leftarrow\boldsymbol{\tau}}$ can be calculated as the Jacobian of $\Omega(\boldsymbol{x}, \boldsymbol{\tau})$ with respect to $\boldsymbol{\tau}$, evaluated at $\boldsymbol{\tau} = \boldsymbol{0}$.

The fifth aspect to be aware of is that the control actions are only two-dimensional (steering and hip torque), whereas the number of balance-relevant muscles and joints is much larger. This simplification can be motivated by the fact that the relevant control input is strongly constrained by the geometry of the bicycle and the rider's position on it. This simplification is specific to balancing a bicycle, and this points to the challenges one may encounter when extending the model to other forms of balance control (e.g., cycling while standing on the

pedals, walking, running, skating, skiing). In principle, the extension is straightforward, as it only requires the EoM for this other form of balancing. However, the challenging part may be the derivation of the EoM, which starts by identifying the balance-relevant joints and selecting the ones that can be actuated. Once the EoM are derived, the linearization and the calculations for the computational system are identical to those for balancing the SDP.

The sixth aspect to be aware of is that the current sensory model is underspecified: it assumes that the sensory input is identical to the state variables $x$ (as implemented by the assumption that the matrix $C$ is the identity matrix) plus some noise. From sensory neurophysiology, it is known that information about the state variables (steering, lower body, and upper body angles and angular rates) must be obtained from the somatosensory and/or the vestibular system, but the details of that knowledge are not yet incorporated in the model.

The seventh aspect to be aware of pertains to the assumption that the motor and the sensor noise are additive, although there is good evidence that motor noise is multiplicative [48, 49]. The advantage of additive over multiplicative noise, is that it is much easier to derive the optimal control actions. For multiplicative noise, optimal control actions were derived by Todorov and colleagues [3, 16], but these are restricted to movements with a finite horizon (e.g., pointing, reaching, throwing, hitting). Keeping balance is an infinite horizon problem (i.e., the cost-to-go functional is an integral from zero to infinity), and this requires mathematical results for which a convenient computational implementation is not yet available [50, 51].

Concluding, I have proposed and evaluated a model for sensorimotor control that is based on the idea that a computational system in the CNS learns and controls an external mechanical system. This control is optimal in the sense of stochastic OFC. The model can balance a bicycle and its rider under realistic conditions and is robust to inaccuracies in the learned noise covariance matrices. It is not robust to inaccuracies in the learned system matrix caused by a misestimation of the speed. The model is a very useful starting point for investigations into human balance control, and there are several ways in which it can be extended to provide a more realistic account.

## Supporting information

**S1 Data. All results on which this paper reports can be reproduced using a set of Matlab live scripts and functions.** This set is documented in the live script BicBalOFC.mlx. (ZIP)

## Author Contributions

**Conceptualization:** Eric Maris.

**Formal analysis:** Eric Maris.

**Investigation:** Eric Maris.

**Software:** Eric Maris.

**Visualization:** Eric Maris.

**Writing – original draft:** Eric Maris.

**Writing – review & editing:** Eric Maris.

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
