## [Decision Letter · Decision Letter 0]

27 Oct 2022

PONE-D-22-23009A bicycle can be balanced by stochastic optimal feedback control but only with accurate speed estimatesPLOS ONE

Dear Dr. Maris,

Thank you for submitting your manuscript to PLOS ONE. After careful consideration, we feel that it has merit but does not fully meet PLOS ONE’s publication criteria as it currently stands. Therefore, we invite you to submit a revised version of the manuscript that addresses the points raised during the review process.

We look forward to receiving your revised manuscript.

Kind regards,

Ning Cai, Ph.D.

Academic Editor

PLOS ONE

Journal Requirements:

Additional Editor Comments :

All two reviewers reach a consensus about the merit of the work. One raises some suggestions for improvement of the presentation. Please prepare a minor revision taking account of the suggestions.

Reviewers' comments:

Reviewer's Responses to Questions

**Comments to the Author**

1. Is the manuscript technically sound, and do the data support the conclusions?

Reviewer #1: Partly

Reviewer #2: Yes

2. Has the statistical analysis been performed appropriately and rigorously? 

Reviewer #1: N/A

Reviewer #2: Yes

3. Have the authors made all data underlying the findings in their manuscript fully available?

Reviewer #1: Yes

Reviewer #2: Yes

4. Is the manuscript presented in an intelligible fashion and written in standard English?

Reviewer #1: Yes

Reviewer #2: Yes

5. Review Comments to the Author

Reviewer #1: The paper overall is interesting. The author replies all the questions from previous reviewers.

1. One major concern is that the paper is too long and contains many known works. The author may consider condensing the work and focus more on the major contributions. The author discussed the human brain and neuro; However, it is not clear to see this in the control design part.

2. Is there any experiment validation of the proposed method.

3. There are still some typos. For instance, in line 26 and line 30, OFC are defined two times.

4. It is not common to see “I” in a research paper. Instead, most time we use “We”.

Reviewer #2: he work is very interesting and difficult. It should be encouraged. There are some minor problems that I hope can be corrected. When the first time a variable appears, it should be written in its full name. For example, BMX on line 310 and LQR on line 552. In line 1467, the units of curvature should be m^{-1}.

6. PLOS authors have the option to publish the peer review history of their article (what does this mean?). If published, this will include your full peer review and any attached files.

Reviewer #1: No

Reviewer #2: **Yes: **Lei Guo

---

## [Author Response · Author response to Decision Letter 0]

11 Nov 2022

Academic editor

>>

I made the following changes to the paper’s style: (1) I placed the Methods section after the Introduction, and (2) I changed the formatting of the headings. 

<<

>>

I ticked the box “The author(s) received no specific funding for this work.” and did not add a Financial Disclosure section because this was not asked for. Should I do something more?

<<

>>

In my Data Availability statement I have indicated that “All relevant data are within the manuscript and its Supporting Information files.”. This paper uses simulations to investigate whether a computational model is plausible for actual bicycle balance control. The documented computer code (Matlab Live scripts) to run these simulations and to extend on them, are provided in the Supporting Information. No experiment has been conducted to test this model, but we make use of empirical information. In fact, we know from experience what turn radiuses are physically possible, and what lean angles are acceptable for a rider. 

<<

>>

I have performed this check, and no papers have been retracted.

<<

Reviewer #1

The paper overall is interesting. The author replies all the questions from previous reviewers.

1. One major concern is that the paper is too long and contains many known works. The author may consider condensing the work and focus more on the major contributions. The author discussed the human brain and neuro; However, it is not clear to see this in the control design part.

>>

It is good to know that the original submission to PLOS Computational Biology was 11 pages shorter than the current version. The length of the current version has increased because I had to reply to the criticism of Reviewer #2 of PLOS Computational Biology. Here, I copy his first three points, all of which ask for more information to be included in the paper: 

“1. The majority of the introduction includes no citations to support the author’s statements. The definition of balance for walking, for example, is well-defined in the literature.

2. There has been significant, serious, and quality research on bicycle dynamics in the past 20 years. While the author does identify a subset of the published works, there also exists significant work examining the human control of the bicycle – most work in this area has been led by Dr. Arend Schwab or Dr. Jason Moore. The current study is not a rigorous scientific study, as it largely ignores the existing knowledge and best approaches for evaluating control models.

3. The author developed his own model of the bicycle rather than using the validated equations of motion for a bicycle. The bicycle the author uses can easily be obtained using the bicycle model of Meijaard et al. 2007 by simply choosing parameter values. The author spends way too much of the paper talking about the development of his own model, which would be unnecessary if he simply utilized the established linearized equations of motion for a bicycle.”

<<

2. Is there any experiment validation of the proposed method.

>>

I did not run an experiment to collect data for testing my model for bicycle balance control. Instead, I used simulations to investigate whether this model is plausible for actual bicycle balance control. For that, I do make use of empirical information. In fact, we know from experience what turn radiuses are physically possible, and what lean angles are acceptable for a rider. Therefore, an empirically plausible model must balance a bicycle without producing turn radiuses that are physically impossible, and/or lean angles that are unacceptable. My model does this. However, it is extremely sensitive to misestimation of the speed, and this is not empirically plausible given what we currently about how the CNS estimates speed. 

<<

3. There are still some typos. For instance, in line 26 and line 30, OFC are defined two times.

>>

Thank you, I corrected those.

<<

4. It is not common to see “I” in a research paper. Instead, most time we use “We”.

>>

I am the sole author of this paper. I did a search on the web, and I found approximately as many fora/web pages that advise “I” for as single author paper as fora/web pages that advise “We”.

<<

Reviewer #2

The work is very interesting and difficult. It should be encouraged. There are some minor problems that I hope can be corrected. When the first time a variable appears, it should be written in its full name. For example, BMX on line 310 and LQR on line 552. In line 1467, the units of curvature should be m^(-1).

>>

Thank you. I added the full names for BMX and LQR. However, I could not find an error w.r.t. the appropriate units of curvature.

<<

---

## [Decision Letter · Decision Letter 1]

25 Nov 2022

A bicycle can be balanced by stochastic optimal feedback control but only with accurate speed estimates

PONE-D-22-23009R1

Dear Dr. Maris,

We’re pleased to inform you that your manuscript has been judged scientifically suitable for publication and will be formally accepted for publication once it meets all outstanding technical requirements.

Kind regards,

Ning Cai, Ph.D.

Academic Editor

PLOS ONE

Additional Editor Comments (optional):

Reviewers' comments:

Reviewer's Responses to Questions

**Comments to the Author**

1. If the authors have adequately addressed your comments raised in a previous round of review and you feel that this manuscript is now acceptable for publication, you may indicate that here to bypass the “Comments to the Author” section, enter your conflict of interest statement in the “Confidential to Editor” section, and submit your "Accept" recommendation.

Reviewer #2: All comments have been addressed

Reviewer #3: All comments have been addressed

2. Is the manuscript technically sound, and do the data support the conclusions?

Reviewer #2: Yes

Reviewer #3: Yes

3. Has the statistical analysis been performed appropriately and rigorously? 

Reviewer #2: N/A

Reviewer #3: Yes

4. Have the authors made all data underlying the findings in their manuscript fully available?

Reviewer #2: Yes

Reviewer #3: Yes

5. Is the manuscript presented in an intelligible fashion and written in standard English?

Reviewer #2: Yes

Reviewer #3: Yes

6. Review Comments to the Author

Reviewer #2: The authors responded properly to my comments. I think the manuscript can be accepted for publication.

Reviewer #3: This paper presents a computational model of this neurobiological component, based on the theory of stochastic optimal feedback control. there is some theoretical value in this article and the quality of article to meet the requirements of the journal.

7. PLOS authors have the option to publish the peer review history of their article (what does this mean?). If published, this will include your full peer review and any attached files.

Reviewer #2: No

Reviewer #3: No

---

## [Editor Report · Acceptance letter]

12 Dec 2022

PONE-D-22-23009R1 

A bicycle can be balanced by stochastic optimal feedback control but only with accurate speed estimates 

Dear Dr. Maris:

I'm pleased to inform you that your manuscript has been deemed suitable for publication in PLOS ONE. Congratulations! Your manuscript is now with our production department. 

Kind regards, 

on behalf of

Dr. Ning Cai 

Section Editor

PLOS ONE